# Scaling Image Geo-Localization to Continent Level

**Philipp Lindenberger** [1]*
plindenbe@ethz.ch

**Paul-Edouard Sarlin** [2]
psarlin.com

**Jan Hosang** [2]
hosang@google.com

**Marc Pollefeys** [1]
mapo@ethz.ch

**Simon Lynen** [2]
slynen@google.com

**Eduard Trulls** [2]
trulls@google.com

[1]ETH Zurich     [2]Google

scaling-geoloc.github.io

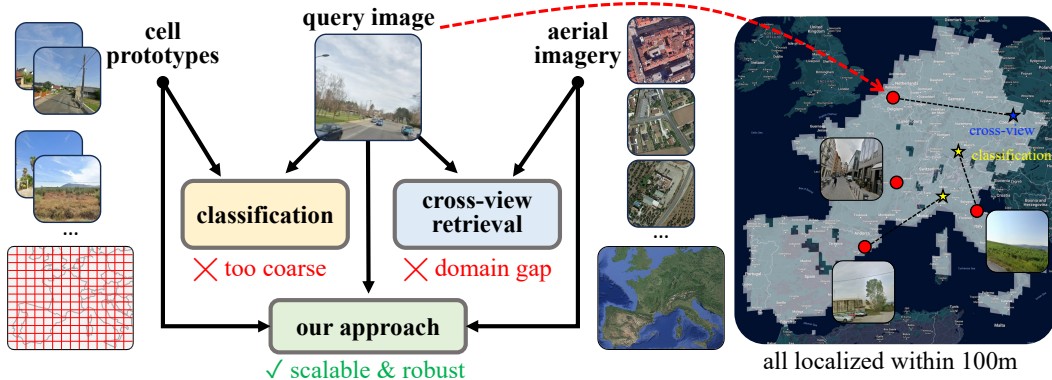

Figure 1: **Large-scale fine-grained geolocalization.** We introduce an approach that can localize a ground-level image within 100 m at the scale of a continent (here Western Europe) by combining the scalability and robustness of classification with the precision of cross-view ground-aerial retrieval. All images shown here are misregistered by either paradigm, but correctly localized by ours.

## Abstract

Determining the precise geographic location of an image at a global scale remains an unsolved challenge. Standard image retrieval techniques are inefficient due to the sheer volume of images (>100M) and fail when coverage is insufficient. Scalable solutions, however, involve a trade-off: global classification typically yields coarse results (10+ kilometers), while cross-view retrieval between ground and aerial imagery suffers from a domain gap and has been primarily studied on smaller regions. This paper introduces a hybrid approach that achieves fine-grained geo-localization across a large geographic expanse the size of a continent. We leverage a proxy classification task during training to learn rich feature representations that implicitly encode precise location information. We combine these learned prototypes with embeddings of aerial imagery to increase robustness to the sparsity of ground-level data. This enables direct, fine-grained retrieval over areas spanning multiple countries. Our extensive evaluation demonstrates that our approach can localize within 200m more than 68% of queries of a dataset covering a large part of Europe. The code is publicly available at scaling-geoloc.github.io.

## 1   Introduction

Pinpointing where in the world an image was taken, down to a scale of meters, remains a challenge in computer vision, especially when scaling beyond city limits [1, 2]. Achieving such fine-grained

---

*Work done during an internship at Google.

39th Conference on Neural Information Processing Systems (NeurIPS 2025).

geolocalization across vast geographic expanses, like entire continents, allows us to localize images without a corresponding GPS tag, such as older and historical images, or images where the EXIF metadata was accidentally stripped, e.g. after image processing. Such technology would be a powerful discriminator to validate images distributed in media, verify images for criminal investigations, or detect AI-generated images. Models trained for large-scale geolocalization need to learn high-level semantics about geographical and cultural patterns from ground views, which could act as a valuable pre-training for remote sensing applications. Furthermore, accurate global priors are often a foundational requirement for more complex downstream tasks, such as 6-DoF positioning based on 3D point clouds or 2D maps [3, 4, 5, 6, 7, 8, 9], which typically require initial estimates within approximately 100 meters.

Current approaches to visual geolocalization force a trade-off between geographic scale and localization precision. **Global classification methods** [10, 11, 1, 12, 2, 13, 14, 15, 16] partition the world into predefined regions (*e.g.*, using administrative boundaries, grid cells, or prominent landmarks) and train classifiers to assign a query image to one of these regions. While highly scalable, these methods are fundamentally limited by the granularity of the partitioning, typically yielding coarse results with errors exceeding 10 km, and are often constrained by the need for sufficient training data per region. On the other hand, fine-grained retrieval approaches aim for higher precision. **Ground-to-ground image retrieval** methods [17, 18, 19, 20, 21, 22, 23, 24] can achieve high accuracy, but struggle to scale to large areas due to the sheer volume of database images. They also suffer from insufficient or uneven geographic coverage of ground-level imagery. **Cross-view retrieval techniques**, matching ground-level queries to aerial or satellite imagery [25, 26, 27, 28, 29], offer better scalability and coverage, but must overcome significant viewpoint and appearance variations and have primarily been studied at sub-country scales. These different streams of research have often evolved independently, lacking a systematic comparison. Consequently, to the best of our knowledge, no existing solution effectively provides both meter-level accuracy and broad, continent-scale applicability.

This paper introduces a novel, hybrid approach designed to bridge the gap between these different research streams, unlocking scalable yet accurate geolocalization by synergizing classification principles with cross-view retrieval (Fig. 1). Our key idea is to leverage a proxy classification task during training, not directly for localization, but to learn *rich, location-specific, ground-view feature prototypes* (Fig. 3). These prototypes implicitly aggregate fine-grained geospatial information visible in ground-level images. Crucially, we then fuse these learned ground prototypes with embeddings of readily available aerial imagery. This mechanism enables efficient and powerful fine-grained cross-view retrieval across vast geographic regions without requiring explicit geometric alignment, dense 3D models, or suffering excessively from the sparse coverage of ground-level training data.

Our contributions are threefold: (i) We propose a novel strategy to **combine learned ground-view prototypes with aerial embeddings** for efficient large-scale yet fine-grained geolocalization. (ii) We demonstrate that fine-grained visual geolocalization is **feasible on a continent-sized region** encompassing multiple countries. Our experiments on a substantial portion of Europe indicate over 68% top-1 recall within 200 m, previously achievable only by city- or regional-scale retrieval systems [30]. (iii) We conduct a **rigorous evaluation** on a benchmark that covers most of western Europe at a much finer scale than previously explored in the literature. We systematically compare our approach against state-of-the-art classification and retrieval methods and provide detailed analyses of model components, such as losses, granularity and backbones, and cross-region generalization capabilities.[2]

By localizing 59.2% of images within 100 m over an area of 284 000 km$^2$, our work demonstrates a path to overcome the long-standing trade-off between precision and scale in visual geolocalization.

## 2 Method

**Problem definition.** Our goal is to localize a query image at ground level $\mathbf{I}^{\mathrm{Q}}$ with high spatial accuracy (~100m) over a very large geographical area (*e.g.* continent-size), without any prior information (*e.g.* GPS). We assume access to a large database of geotagged ground-level images $\{\mathbf{I}_i^{\mathrm{G}}\}$ and tiled overhead (aerial or satellite) images $\{\mathbf{I}_j^{\mathrm{A}}\}$. Modern solutions [17, 31, 24, 30, 22] encode the query image into a $D$-dimensional embedding using a deep neural network $\Phi : \mathbb{R}^{H \times W \times 3} \rightarrow \mathbb{R}^D$ and localize it via similarity search against embeddings inferred from the database images.

---

[2]Analytical use of StreetView imagery was done with special permission from Google.

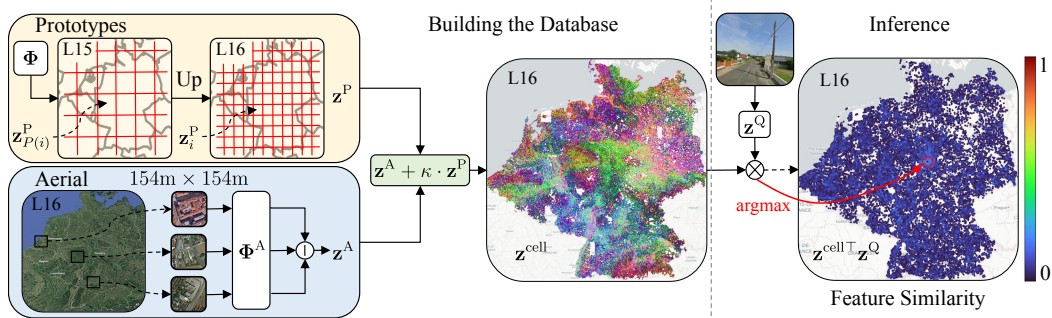

Figure 2: **Inference pipeline.** The prototypes $\mathbf{z}^{\mathrm{P}}$ are extracted from the model weights $\Phi$ and upsampled to the target resolution using the S2Cell hierarchy. Aerial tiles roughly covering the cell are encoded using the aerial encoder $\Phi^{\mathrm{A}}$ and concatenated. Both databases are combined per-cell using the calibration factor $\kappa$, resulting in the final database of cell codes $\mathbf{z}^{\mathrm{cell}}$. During inference (right), we extract the embedding of a query image $\mathbf{z}^{\mathrm{Q}}$ with $\Phi^{\mathrm{G}}$ and we compute the similarity to all cell codes $\{\mathbf{z}_j^{\mathrm{cell}\top}\mathbf{z}^{\mathrm{Q}}\}$. The estimated location is the cell with the highest similarity.

**The precision/scaling trade-off.** Retrieval-based (VPR) approaches have been studied extensively but are hard to scale, given their inherent limitation that every query image must have visual overlap with at least one database image to be localized. With their narrow field-of-view (FOV), this requires an image database that densely covers the environment. As such, both storing the database embeddings and querying them quickly becomes prohibitively expensive: a VPR method with performance comparable to our approach would require storing embeddings for 470 million images for our largest dataset (`EuropeWest`, Fig. 1). They can also be difficult to train, as contrastive learning is sensitive to the sampling of positive and negative pairs, which becomes difficult (and more important) at scale.

One way to approach scaling is via classification, *i.e.*, partitioning the space into disjoint classes, such as cells in a regular grid, whose prototypes summarize *all images in the area*. This reduces the size of the database and eschews the need for negative mining, as each example can be contrasted to *all* prototypes—at the cost of accuracy, which is bounded by the granularity of the partitioning. Finer partitions increase the number of prototypes, which is bounded by the available memory at training time, and additionally impair convergence and generalization as each class is represented by fewer examples. Additionally, at inference time, these methods are limited to areas covered at training time. Despite these limitations, classification-based methods are popular—in fact, VPR methods often train classification losses as a proxy task [18, 22] and revert to similarity search at inference time.

An alternative approach is cross-view localization using overhead imagery, which is inherently more scalable, as one image tile can cover (and thus summarize) a larger area, reducing the size of the database. Overhead imagery is also easier to acquire, and often readily available from public sources—many rural roads are not covered by any ground-level imagery available on the internet. These approaches however suffer from a large domain gap between database and query images, as vertical structures like building facades are usually not visible in near-nadir imagery.

There is thus a clear trade-off between precision (*i.e.*, each image is represented in the database) and scalability (*i.e.*, classification over larger regions). These different research streams are rarely compared or studied in combination. In this paper we combine their strengths. We provide more details on previous work and how it relates to our method in Section 4.

**Our solution: Combine cell prototypes and cross-view retrieval (Fig. 2).** We propose to perform retrieval over *cell codes* that *combine classification prototypes and embeddings of aerial tiles*: a surprisingly simple formulation that has, to our knowledge, not been explored until now. We thus learn $l2$-normalized class prototypes $\{\mathbf{z}_j^{\mathrm{P}}\}$ via a classification proxy task. For partitioning we rely on the S2 cell library [32], which defines a hierarchy over the Earth: a cell of a given level $L{=}X$ can be split into 4 cells of level $L{=}X{+}1$ (higher is finer). We use cells at fixed levels, with cells being roughly (but not exactly) equal. While previous works have used adaptive cell sizes to ensure that each cell includes enough training examples, we find that this degrades accuracy in rural areas with a lower density of database images. Instead, we rely on aerial information to constrain such cells. Administrative regions, as used in previous works [1, 33], are far too coarse for our use-case.

We define aerial tiles $\mathbf{I}_i^A$ of fixed size and ground sampling distance, centered at each cell. Aerial tiles are encoded by a neural network $\Phi^A$ into $l2$-normalized embeddings $\mathbf{z}_i^A$. We typically use cell codes at $L{=}15$ (average edge length $\sim$281 m), which is the upper limit we can store at training time, and aerial tiles at $L{=}16$ (average edge length $\sim$140 m). Prototypes and aerial embeddings are then *combined into cell codes* at the finest granularity (typically level $L{=}16$) in a straightforward manner: $\mathbf{z}_i^{\text{cell}} = \kappa \cdot \mathbf{z}_{P(i)}^P + \mathbf{z}_i^A$, where $P(i)$ defines the index of the cell at $L'$ that is parent of cell $i$ at level $L$. The calibration factor $\kappa \in \mathbb{R}$ is introduced to account for the empirical observation that the similarities from queries to prototypes are generally smaller in magnitude than to the aerial embeddings—we attribute this to the different granularity levels: cell code prototypes encode a larger area (and more images), so the deviation of their embeddings is larger. Empirically we set $\kappa$ to match the average top-1 similarities over the training set. We then embed a given query image into $\mathbf{z}^Q$ using the ground-level encoder $\Phi^G$ and search for its nearest neighbors in the database $\{\mathbf{z}_i^{\text{cell}}\}$.

Both aerial and ground-level encoders $\Phi^A$ and $\Phi^G$ use the same architecture, but with different weights. The aggregation of patch embeddings into a single vector is performed with the optimal transport head introduced in SALAD [24].

**Training recipe (Fig. 3).** We use database ground-level images as "simulated" queries. Each training example includes a ground-level query and an associated aerial tile. The aerial tile is centered at the location of the query, with a random rotation and translation offset, for data augmentation. We train the aerial and ground-level encoders $\Phi^A$ and $\Phi^G$, as well as the prototypes $\{\mathbf{z}_i^P\}$, jointly. Prototypes learn to summarize a given area based on the training with corresponding queries.

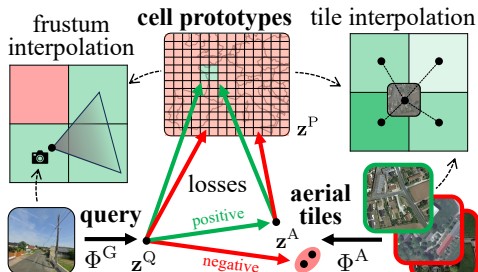

Figure 3: **Supervision:** We train query, aerial, and prototype embeddings, $\mathbf{z}^Q$, $\mathbf{z}^A$ and $\mathbf{z}^P$, to be similar for corresponding locations and different otherwise. We interpolate prototypes to account for the coarseness of their cells.

We follow a contrastive learning formulation, in which we want ground-level, aerial, and prototype embeddings to be similar when they correspond to the same spatial location and distinct otherwise. These three elements form a triangle with three constraints, one along each edge: ground-prototype, ground-aerial, aerial-prototype. (i) Each query is contrasted to all prototypes, such that they learn to encode features that represent what is visible from corresponding ground-level images—*e.g.* facades, bridges, small details, elements under vegetation. (ii) Each query is also contrasted to all aerial tiles that are in the batch, such that ground-level and aerial encoders learn to embed similar information. (iii) Finally, each aerial tile is contrasted to all prototypes, such that aerial embeddings are trained to be globally discriminative, eschewing the need for hard negative mining. Because we build a separate aerial database, we detach the prototype gradients in this edge, forcing the network to only aggregate ground-view information in the prototypes. We ablate these loss terms in detail in the supplementary material, Table 9.

**Loss formulation.** The positive and negative terms of each of the three constraints are combined into a multi-similarity loss [34]. We define the total loss as $\mathcal{L} = \sum_{i \in \mathcal{B}}(\mathcal{L}_i^{\text{pos}} + \mathcal{L}_i^{\text{neg}})$, where the positive term is computed for each query $i$ in the batch $\mathcal{B}$ as

$$\mathcal{L}_i^{\text{pos}} = \frac{1}{\alpha} \log \left( 1 + \gamma(\mathbf{z}_i^{Q\top}\mathbf{z}_i^A) + \gamma(\mathbf{z}_i^{Q\top}\mathbf{z}_i^P) + \gamma(\mathbf{z}_i^{A\top}\mathbf{z}_i^P) \right) \quad , \tag{1}$$

and the negative term is

$$\mathcal{L}_i^{\text{neg}} = \frac{1}{\beta} \log \left( 1 + \sum_{j \in \mathcal{B}\setminus\{i\}} \left( \delta(\mathbf{z}_i^{Q\top}\mathbf{z}_j^A) + \delta(\mathbf{z}_i^{A\top}\mathbf{z}_j^Q) \right) + \sum_{j \in \mathcal{N}(i)} \left( \delta(\mathbf{z}_i^{Q\top}\mathbf{z}_j^P) + \delta(\mathbf{z}_i^{A\top}\mathbf{z}_j^P) \right) \right) . \tag{2}$$

Here $\mathcal{N}(i)$ denotes the indices of all prototype cells whose spatial distance to the query $i$ is larger than a threshold. The functions $\gamma, \delta : \mathbb{R} \to \mathbb{R}$ enforce positivity with scale and bias hyper-parameters $\alpha, \beta, \lambda \in \mathbb{R}$ and are defined as $\gamma(s) = e^{-\alpha(s-\lambda)}$, and $\delta(s) = e^{\beta(s-\lambda)}$.

Prototypes are associated with discrete cells but ground-level and aerial images are sampled continuously through space. Those located at cell boundaries should thus be treated carefully to avoid artifacts that can impair the training. We linearly interpolate each positive prototype $\mathbf{z}_i^P$ with its

Table 1: **Recall on** `BEDENL`. (a) In traditional retrieval methods, the database size corresponds to the number of ground-level images in the training set: they perform best yet are often intractable (the state-of-the-art SALAD generates a 5 TB database). (b) Cross-view retrieval is faster but lags in performance. (c) Classification produces smaller databases and suffers less from domain gap, but s bound by device memory at training time ($L{=}15 \Rightarrow$ coarser cells). (d) Our hybrid approach combines the benefits of (b-c) and performs comparably to (a) with a 30-60$\times$ smaller database.

| | Eval | Method | cell level | Recall@$K$@200m | | | Recall@$K$@1km | | | database | | |
| --- | --- | --- | --- | --- | --- | --- | --- | --- | --- | --- | --- | --- |
| | | | | $K{=}1$ | $K{=}5$ | $K{=}100$ | $K{=}1$ | $K{=}5$ | $K{=}100$ | size | elems. | dim. |
| (a) | ground retrieval | SALAD [24] | | OOM | OOM | OOM | OOM | OOM | OOM | 5.1 TB | 150M | 8448 |
| | | Ours (cell prototypes) | 16 | 56.3 | 66.2 | 80.4 | 57.5 | 66.9 | 81.3 | | | |
| | | Ours (full) | | **64.0** | **73.8** | **85.7** | **65.3** | **74.5** | **86.5** | 1.3 TB | 150M | 2176 |
| (b) | aerial retrieval | Fervers *et al.* [30] | | 33.3 | 48.9 | 74.2 | 36.2 | 51.3 | 75.5 | | | |
| | | + negative mining | | 36.5 | 51.1 | 75.5 | 39.3 | 53.4 | 76.5 | 42 GB | 4.8M | 2176 |
| | | SALAD [24]-Aerial | 16 | 39.2 | 53.6 | 74.1 | 42.6 | 56.2 | 77.0 | | | |
| | | Ours (full) | | 49.7 | 63.2 | 81.0 | 52.0 | 64.8 | 82.7 | | | |
| (c) | cell prototypes | CosFace loss [38] | | 7.4 | 13.3 | 19.4 | 10.3 | 18.3 | 26.9 | | | |
| | | Hierarchical loss [2] | | 8.1 | 14.5 | 27.8 | 14.0 | 23.2 | 43.2 | | | |
| | | Haversine loss [1] | 15 | 10.2 | 19.5 | 36.2 | 17.8 | 25.7 | 42.0 | 18 GB | 2M | 2176 |
| | | Ours (cell prototypes) | | 47.3 | 59.4 | 74.7 | 49.9 | 60.9 | 76.9 | | | |
| | | Ours (full) | | 57.1 | 68.6 | 81.8 | 59.5 | 69.9 | 83.0 | | | |
| (d) | hybrid | Ours (smaller dim.) | | 58.0 | 69.3 | 83.8 | 57.2 | 67.6 | 81.3 | 21 GB | 4.8M | 1088 |
| | | Ours (full) | 16 | **60.3** | **71.6** | **85.6** | **62.0** | **72.4** | **86.4** | 42 GB | 4.8M | 2176 |
| | | Ours (DINOv3-L) | | 70.2 | 79.0 | 89.9 | 71.8 | 79.5 | 90.4 | 42 GB | 4.8M | 2176 |

neighbors, with weights computed based on either (i) the overlap between the respective prototype cells and the camera frustum of the query, which is defined as a 2D triangle with 50 m depth, or (ii) the distance between the centers of the aerial tile and of the 4 closest prototype cells (Fig. 3).

We empirically observe that this multi-similarity loss outperforms cross-entropy losses like In-foNCE [35] and DCL [36]. It constrains the absolute similarity instead of the relative similarities. We hypothesize this prevents images that cannot be localized, *e.g.* because of occlusion or lack of distinctive features, from dominating the loss, as the gradient of their negative term remains small.

**Pushing the boundaries of scalability.** When scaling up classification models to a very large number of classes, the devil is often in the details—our largest model is trained with 7M cell codes (Table 2). We highlight our most important learnings. The size of the prototypes during training is the limiting factor when naively replicating them across devices. For example, with dimension $D{=}2176$, only ~250k prototypes can fit on each device. As a reference, this roughly covers Belgium at $L{=}15$. To alleviate this, we uniformly shard the $N$ prototypes across all $d$ devices, such that each holds only $N/d$ of them. The backbone is replicated across devices for a fast forward pass. We gather the image embeddings over all devices and compute the image-prototype similarity per-device. We then compute a subset of the negative sum (Eq. (2)) locally and broadcast it to all devices. This minimizes data transfers and thus keeps the training fast. It is faster than replicating the entire forward pass, which transfers prototype gradients between devices. We train with 128 16GB TPUv2 [37].

## 3 Experiments

**Datasets.** Our model ingests both overhead and ground-level imagery. Publicly available ground-level datasets are not of sufficient scale or density for our purposes. We use Google StreetView imagery, captured by six rolling-shutter, fish-eye cameras mounted on cars. StreetView rigs all have similar camera models and angles relative to the road, so we found data augmentation crucial to prevent overfitting: we stitch StreetView images into panoramas and sample crops of 224$\times$224 pixels with a pinhole camera model and random roll, pitch, yaw, and FOV. In order to sample a consistent number of images per region, we select panoramas via farthest point sampling over both space and capture time in order to maximize spatial and temporal coverage. We consider a maximum of 120 panos per $L{=}14$ S2 cell [32], enforce that panos are least 40 m apart to prevent oversampling, and skip cells that contain less than 5 panos. We then generate 4 image crops per pano. We use sequences captured in year 2023 *for evaluation only* and sequences from the remaining years 2017–2024 for training.

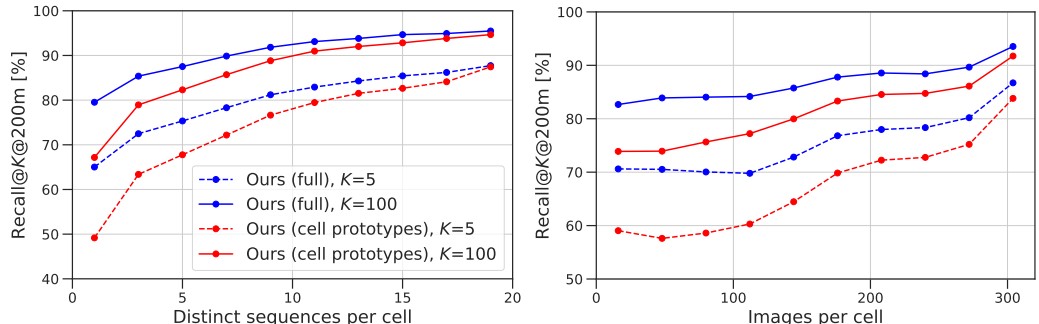

Figure 4: **Impact of the density of the training data.** We slice the recall@$K$@200m on `EuropeWest` (Table 3) by the temporal (left) and spatial (right) density of StreetView images within $L$=15 cells. We compare our full (hybrid) model (**blue**) with one relying only on ground-level images (**red**). The aerial embeddings help improve the accuracy especially when ground-level data is sparse.

For overhead assets, we use both aerial (captured by planes) and satellite images. We pick the highest-resolution asset available and sample 256×256 px tiles at a 60 cm/px resolution. For training, we sample tiles with a maximum offset of 80 m w.r.t. the ground-level images, and randomly rotate them. For inference, we sample north-aligned tiles centered at the S2 cells.

We consider two geographical regions: `BEDENL`, consisting of three countries (BE, DE, NL), and a larger superset `EuropeWest`, with ten countries comprising most of western Europe (PT, ES, IT, AT, CH, DE, FR, BE, NL, CZ). We use `BEDENL` for most experiments and ablation studies and `EuropeWest` to demonstrate the scalability to larger regions.

We evaluate generalization capabilities on three other datasets, see Table 2: (i) `UK+IE`: Two additional countries (UK, IE) for which we build the database using only overhead images and query with StreetView images *without re-training*. (ii) `Trekker`: StreetView images from backpacks [39], worn by walking operators, as queries for models trained on `BEDENL`. They have a different spatial distribution (*i.e.*, pedestrian-centric) and suffer from occlusions as the devices are often close to walls.

(iii) `GoogleUrban`: Images captured by consumer phones, covering 5 cities included in `BEDENL`, part of a proprietary dataset previously used to evaluate localization algorithms in urban settings [5] and in the 2022 Kaggle Image Matching Challenge [40].

**Metrics.** At inference time we extract embeddings for a new (unseen) set of images and find the database embeddings with highest similarity. The database can be be built from ground-level images ("ground retrieval" in Table 1, *i.e.*, VPR), aerial tiles ("aerial retrieval"), cell codes learned via classification, or a combination of aerial and cell code retrieval. Note that for models trained for classification, similarity search in feature space and top-$K$ classification are equivalent. We evaluate performance in terms of localization recall at different spatial thresholds and for different number of nearest neighbors $K$. The accuracy

Table 2: **Datasets.** Number of ground-level images and S2 cells (at $L$=15) and approximate area covered (km$^2$).

| Dataset | Training | | | Evaluation | | |
|---|---|---|---|---|---|---|
| | Images | Cells | Area | Images | Cells | Area |
| `BEDENL` | 150M | 2.0M | 139k | 1.5M | 378k | 25.7k |
| `EuropeWest` | 470M | 7M | 433k | 4.5M | 1.2M | 69.7k |
| `UK+IE` | N/A | N/A | N/A | 1.2M | 194k | 16k |
| `Trekker` | N/A | N/A | N/A | 130K | 4.3k | 1.5k |
| `GoogleUrban` | N/A | N/A | N/A | 767k | 3.1k | 1.1k |

Table 3: **Left: Recall on `EuropeWest`.** Note that VPR is *infeasible* at this scale (470M images). **Right: Recall on `UK+IE` for the model trained on `EuropeWest`.** The database is built from *aerial tiles only*, without retraining.

| Method | `EuropeWest` Recall@$K$@200m | | | Cross-Area (`UK+IE`) Recall@$K$@200m | | |
|---|---|---|---|---|---|---|
| | $K$=1 | $K$=5 | $K$=100 | $K$=1 | $K$=5 | $K$=100 |
| Fervers *et al.* [30] | 32.6 | 47.6 | 72.1 | 11.6 | 19.5 | 39.3 |
| Ours (prototypes) | 46.4 | 58.4 | 72.7 | N/A | N/A | N/A |
| Ours (full) | **57.5** | **69.4** | **84.3** | **18.4** | **28.1** | **47.2** |
| Ours (DINOv3-L) | 68.7 | 78.1 | 89.1 | 27.4 | 38.2 | 58.6 |

of classification-based methods is limited by their granularity level, *i.e.*, $L$=15 (average cell edge length 281 m), whereas our aerial and hybrid models work at $L$=16 (average cell edge length 140 m). Given space constraints and for a fair comparison we report recall at 200 m and 1 km, irrespective of cell granularity. Refer to the appendix for localization results at 100 m on the `EuropeWest` dataset.

**Evaluation—Scaling to a continent (Table 1 & 3–left).** We conduct large-scale experiments over multiple countries in Europe. **Setup:** Because of the large compute requirements, we do hyper-parameter search on BEDENL and then train a larger model on EuropeWest. **Baselines:** For *cross-view retrieval*, we re-train a model similar to that of Fervers *et al.* [30], but with the same granularity and aerial tile size used by our method. To strengthen this baseline, we perform offline hard-negative mining (using k-NN) between aerial embeddings once during training. We also adapt SALAD [24], a state-of-the-art ground-based retrieval model [24], to cross-view retrieval with the multi-similarity loss [34]. For *classification*, we train two baselines with variants of the cross-entropy loss: a

Table 4: **Generalization to pedestrian viewpoints.** We report localization recall on the urban Trekker dataset.

| Method | Recall@$K$@200m | | |
|---|---|---|---|
| | $K$=1 | $K$=5 | $K$=100 |
| Ours (VPR) | 21.5 | 29.9 | 46.1 |
| Fervers *et al.* [30] | 11.1 | 20.6 | 47.1 |
| SALAD [24]-Aerial | 13.4 | 23.1 | 46.0 |
| Ours (prototypes) | 15.5 | 24.8 | 44.3 |
| Ours (full) | **18.7** | **29.3** | **51.3** |
| Ours (full/BEDENL+) | **30.3** | **44.2** | **63.5** |

hierarchical loss from OSV-5M [2] and with the smoothing introduced by PIGEON [1], based on the Haversine distance between training images and cell centers. We provide additional details on the baselines and a full ablation of loss terms in the supplementary. For completeness, we also report results for traditional *ground-based retrieval* for BEDENL in Table 1 —for EuropeWest (Table 3) the database would simply be too large. **Results:** On BEDENL, classic image-to-image retrieval exhibits high accuracy but is computationally infeasible. Cross-view retrieval drastically reduces the size of the database, but the domain gap impairs recall. Classification achieves higher recall, but the sparsity and diversity of the data hinders the embedding averaging in a cell. Our approach combines the strengths of both. A larger backbone, DINOv3-L [41], yields another substantial performance gain.

**Evaluation—Cross-area generalization (Table 3–right).** A fundamental shortcoming of geolocalization methods based on classification techniques is that they require retraining when faced with new data. We show that our approach can generalize to completely unseen areas by building the database *using only aerial imagery*. Despite a drop in performance, our approach remains applicable and can recover over half the queries at $K$=100. Note that given ground-level images, we could also do VPR with their image embeddings, but this would require indexing 109M images for UK+IE. Here we use 2.8M aerial tiles.

**Evaluation–Cross-domain generalization (Table 4).** We evaluate our approach on queries from the Trekker dataset, which contains Google StreetView images taken from the vantage point of pedestrians. In addition to drastic viewpoint differences (road *vs* sidewalk), many images are unlocalizable, as cameras often closely face building facades. We feed them to our model trained on BEDENL, without any fine-tuning. The drop in performance is primarily explained by three factors: (i) viewpoint difference, (ii) unlocalizable images, and (iii) this dataset is only available for urban centers, while our training recipe prioritizes good coverage of *all* cells, the majority being in rural areas. To validate (iii), we train a new model on

Table 5: **Impact of cell size $L$ and feature dimensionality $D$.** We report recall on BEDENL. $N$ is the number of S2 cells.

| $L$ | $D$ | Recall@$K$@200m | | | $N$ |
|---|---|---|---|---|---|
| | | $K$=1 | $K$=5 | $K$=10 | |
| 12 | 2048+128 | 37.8 | 50.7 | 70.3 | **86K** |
| 14 | 2048+128 | 50.2 | 62.8 | 78.9 | 760K |
| 15 | 2048+128 | 60.3 | 71.6 | 85.6 | 2.0M |
| 16 | 2048+128 | **63.3** | **73.8** | **87.1** | 4.8M |
| 13 | 8192+256 | 46.1 | 59.3 | 76.2 | 278K |
| 14 | 4096+128 | 52.8 | 64.8 | 80.6 | 760K |
| 15 | **1024+64** | 58.0 | 69.3 | 83.8 | 2.0M |
| 16 | **1024+64** | 60.7 | 72.5 | 86.4 | 4.8M |

Table 6: **Impact of the frustum and cell interpolation.** We compare them to nearest neighbour sampling, on BEDENL.

| Ground | Aerial | Recall@$K$@200m | | |
|---|---|---|---|---|
| | | $K$=1 | $K$=5 | $K$=100 |
| NN | NN | 57.8 | 69.8 | 84.6 |
| Frustum | NN | 58.9 | 70.5 | 85.0 |
| NN | Interp. | 58.5 | 70.3 | 84.9 |
| Interp. | Interp. | 57.7 | 69.5 | 84.3 |
| Frustum | Interp. | **60.3** | **71.6** | **85.6** |

BEDENL+, a dataset covering the same areas in BEDENL but sampling a number of images per cell proportional to the number of training sequences, instead of uniformly. This greatly improves performance (as urban areas contain more sequences) and highlights the trade-off between localization in urban centers and rural areas, dominated by roads.

Table 7: **Ablation of encoders on** `BEDENL`. Larger models and input images yield a better localization. The initialization matters too.

| Image size (px) | Encoder size & init. | Recall@$K$@200m | | |
|---|---|---|---|---|
| | | $K$=1 | $K$=5 | $K$=100 |
| | DINOv2-S14 | 57.0 | 67.6 | 82.0 |
| 224 | SigLIP 2-B16 | 57.4 | 64.6 | 83.2 |
| | iBOT-B16 | 60.3 | 71.6 | 85.6 |
| | DINOv2-B14 | 63.5 | 73.4 | 86.2 |
| | DINOv3-B16 | 64.1 | 74.1 | 86.8 |
| | DINOv2-L14 | 69.7 | 78.5 | 89.5 |
| | DINOv3-L16 | **70.2** | **79.0** | **89.9** |
| 448 | DINOv3-L16 | **76.1** | **84.1** | **93.0** |

Table 8: **Generalization to phone images on** `GoogleUrban`. Fine-tuning with different image resolutions helps generalization.

| Image size (px) test | training | Encoder | Recall@$K$@200m | | |
|---|---|---|---|---|---|
| | | | $K$=1 | $K$=5 | $K$=100 |
| 224 | 224 | DINOv2-L14 | 24.5 | 39.3 | 70.8 |
| | | DINOv3-L16 | 27.5 | 42.7 | 73.2 |
| | 224 & 448 | DINOv2-L14 | 23.4 | 38.1 | 70.0 |
| | | DINOv3-L16 | 30.3 | 46.1 | 76.0 |
| 448 | 448 | DINOv2-L14 | 29.4 | 45.6 | 75.6 |
| | | DINOv3-L16 | **42.7** | **59.1** | **83.3** |
| | 224 & 448 | DINOv2-L14 | 27.1 | 43.2 | 75.4 |
| | | DINOv3-L16 | 38.8 | 55.2 | 81.6 |

**Ablation—Granularity (Table 5).** The resolution of the grid at cell level $L$ and the feature dimensionality $D$ have a large impact on accuracy. We use our 'hybrid' model and report recall at 200 m. For a given $D$, coarser grids yield lower recall, especially at $L \leq 14$. Coarse cells, often employed in the literature [2, 1, 10], need to encode a quadratically growing area, thus increasing the visual diversity within a class. While some works aim to alleviate this problem by finding semantically meaningful clusters [1], there is no guarantee that these features are observable in each image. At constant database size, using more compact features with a higher-resolution grid is thus a better trade-off between scalability and accuracy.

**Ablation—Border interpolation (Table 6).** We study strategies to align query and aerial embeddings to their corresponding prototypes. The simplest one selects the cell nearest to the query or aerial tile. We compare this to bilinear interpolation based on frustum/tile overlap and for queries, to selecting all cells covered by their 2D camera frustum. We report the recall for our best, 'hybrid' model at 200 m. The results show that the interpolation is always beneficial for aerial tiles. On the other hand, it impairs recall for queries, likely because it does not consider how far the scene is visible from the ground. Surprisingly, selecting all overlapping cells works best.

**Ablation—Backbones and image resolution (Table 7 & Table 8).** We ablate the ViT vision foundation models iBOT [42] (default), DINOv2 [43], SigLIP 2 [44] and DINOv3 [41], and their variants. On `BEDENL`, Table 7, larger backbones yield substantial improvements. DINOv3 [41] generally works best, while finetuning and evaluation on larger images (448x448 px) yields again significant improvements. In Table 8, we study the model's generalization capabilities to casual images from smartphones in urban areas, `GoogleUrban`. Our model with DINOv3-L16 [41] backbone, finetuned and evaluated at 448px images, achieves almost 60% top-5 recall, suggesting strong generalization to casual images. Note that, for generalization to non-square images, we augment the training images with random zero padding on either the outer rows or columns.

**Qualitative results.** Figure 5 shows the prototypes learned from the `EuropeWest` dataset, visualized with PCA, and the localization errors of our test set. Figure 6 shows examples of queries.

**Implementation details.** We train our models with 64 examples per batch per device (8192 examples in total) for 200k steps (~3 epochs on `EuropeWest`), with ~1 s per step and a total time of 2.5 days. We use the Adam [45] optimizer with learning rates of 0.003 for the encoders and 0.01 for the prototypes, both annealed to $10^{-6}$ by the end of training using a cosine schedule. Unless stated otherwise, we use Vision Transformers [46] (B16) initialized with the iBOT [42] weights, and scale the best setups to larger models. During training we randomly drop layers with a probability of 0.1. The SALAD head [24] has 32 64-D clusters and a 128-D class token. All embeddings are $l$2-normalized. The loss is parameterized by $\alpha$=0.2, $\beta$=100, and $\lambda$=0.2.

## 4 Related Works

**Visual Place Recognition (VPR)** approaches image localization as matching a query image against a large database of geo-tagged ground-level images, typically via pre-extracted image features designed to be robust to changes in viewpoint and illumination and to occlusions [47, 48, 49, 50, 51, 52, 53, 31]. In its simplest form, the query is then assigned the location of the closest image in the database. While

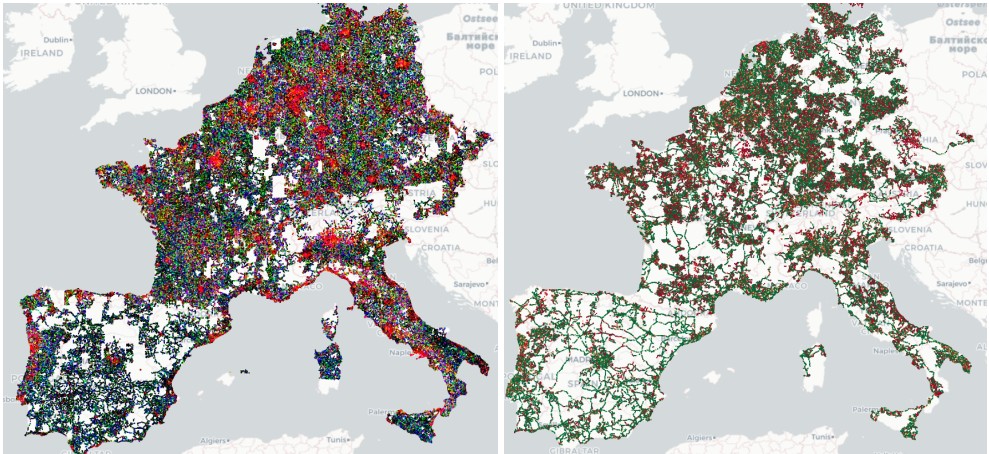

Figure 5: **Left:** PCA visualization of the learned prototypes, which appear in different colors for *e.g.*, urban, forested, or coastal areas. The high-frequency noise suggests that they also encode local distinctive information. **Right:** Test queries that are successfully localized (●) are uniformly distributed over the map, while failures (●) are prevalent in rural areas, where training data is sparser.

early papers relied on handcrafted features [54, 55, 56], the main focus of modern VPR is learning discriminative and compact representations. CosPlace [18] introduced a city-sized dataset and showed that previous methods fail to scale to it, proposing to learn descriptors for retrieval with a proxy classification loss to bypass the expensive mining required by contrastive learning. Vo *et al*. [57] similarly concluded that the best features for retrieval are trained via classification. EigenPlaces [19] defined classes as to enforce viewpoint invariance in the learned features. TransVPR [58] signaled a move towards ViTs [46] and self-attention [59]. MixVPR [60] proposed an MLP-based feature mixer to aggregate features from off-the-shelf foundation models, while AnyLoc [20] explored unsupervised aggregation techniques [61, 62, 63, 50]. SALAD [24] proposed an aggregation strategy based on optimal transport with DINOv2 features and was subsequently improved with a mining strategy that accounts for geographic distance [23]. MegaLoc [22] trained a single retrieval model on five large datasets and showed it outperforms most previous models. Other research topics within this scope include geometric verification and re-ranking [4, 64, 53, 65, 66, 67, 21] and uncertainty estimation [68, 69, 70, 71, 72]. Recently, MeshVPR [73] combined global features for retrieval with a visual localization step based on local features and dense 3D textured meshes.

**Cross-View localization** compares ground-level images to overhead views, such as satellite imagery. In practice, ground-level imagery is often too scarce to ensure global coverage. Early efforts relied on warping image semantics to the overhead reference frame before matching [74, 75, 76, 77]. This topic gained traction with the advent of deep learning [78, 79]. Liu *et al*. [80] learned orientation-aware features by explicitly encoding per-pixel orientation information. Shi *et al*. [25] used polar transforms to warp aerial images into panoramas, with an attention mechanism to alleviate distortions, while [81] used feature transport for domain transfer across views. Ye *et al*. [27] proposed a technique to convert panorama images into overhead views, while also directly matching unwarped panoramas to satellite images. ConGeo [28] enhanced robustness on non-north-aligned panoramas and variable fields of view. Zhang *et al*. [26] used synthetic augmentations to benchmark cross-view localization methods against weather, blur, or image compression. In a different direction, OrienterNet [7] matched ground-level images to overhead semantic maps and SNAP [9] learned neural maps directly from images—both use very small tiles and require GPS priors. Recently, Fervers *et al*. [30] demonstrated the applicability of such methods to areas the size of the state of Massachusetts, with ~60% accuracy at 50m. Their approach partitions regions into cells that factor in spherical distortions and combines multiple scales of overhead images. Our approach shows higher accuracy over much larger areas.

**Global geolocalization** focuses on scaling up to much larger areas, up to Earth-scale. While they employ different design paradigms, classification techniques are most common. Early works such as IM2GPS [47] extracted simple image features from a database of 6M geo-tagged images and used retrieval for inference—our largest database consists of 470M images, rendering retrieval prohibitive. PlaNet [10] partitioned the Earth's surface into S2 cells [32] into which images are classified, but suffers from limited precision, with only 26k cells of variable size. Clark *et al*. [13] introduced learned latent features over hierarchical S2 cells, but remain limited to a similar number of cells at the

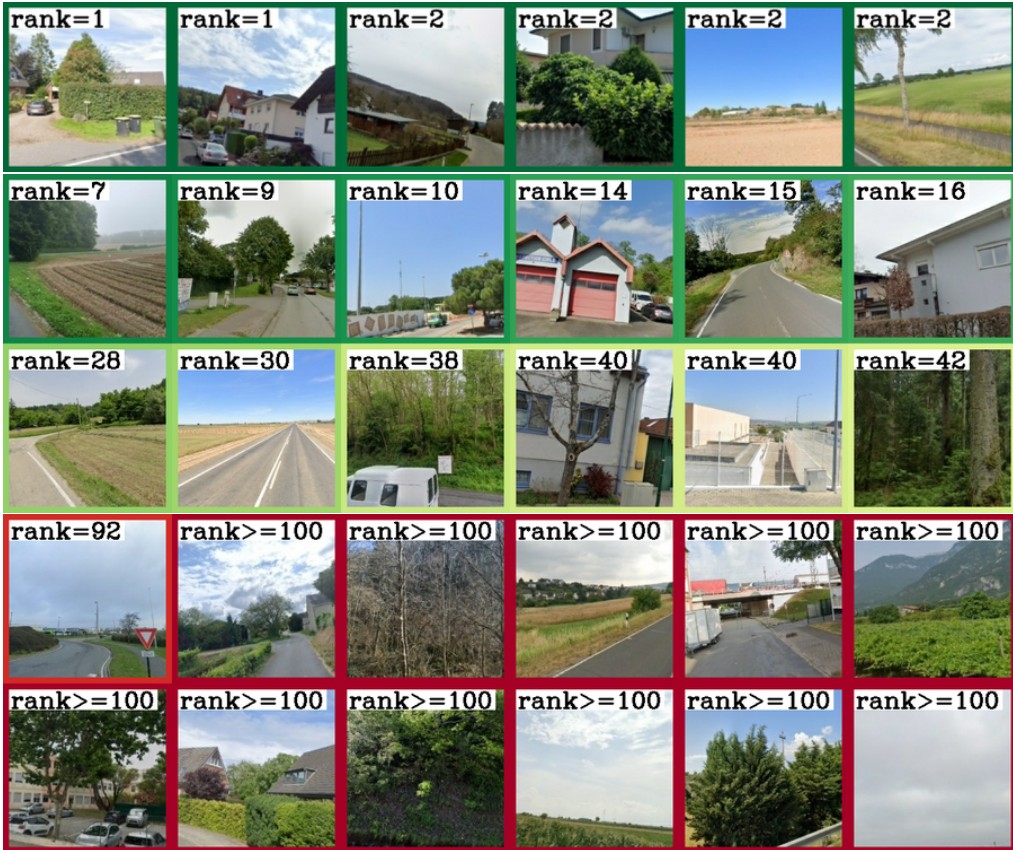

Figure 6: **Qualitative examples.** Localization examples of easy, medium, and difficult cases, along with their rank (the position of the first cell within 200m in the sorted database list according to descriptor similarity). A larger rank corresponds to a lower localization accuracy.

finest level. CPlaNet [11] applied combinatorial partitioning techniques to increase this number to 2.8M—our largest model can accommodate 18M cells. Translocator [82] used RGB images and their segmentation maps as inputs to increase robustness against weather or illumination changes. OSV-5M [2] introduced a global-scale dataset of open-sourced ground-level images and evaluated different image encoders, pretraining sets, and supervision schemes, including regression, classification, and a hybrid approach, as well as contrastive objectives at semantic partitions such as administrative regions. It relies on a strict spatial split, where images in the test set are at least 1 km away from any image in the training set, and evaluates the classification accuracy at country, region, and city levels. OSV-5M aims to learn geographical features without explicitly encoding appearance, while our goal is different—we wish to *summarize appearance*, which is needed for fine-grained localization, and thus use much smaller cells and a temporal split for evaluation. PIGEON [1] introduced semantic cells and a regularization to relate adjacent cells to each other. Other works explored contrastive learning to align images to GPS locations or text captions [12, 83, 1]. In a different direction, Dufour *et al*. [16] explored a generative approach based on diffusion.

## 5    Conclusion

We address the challenge of high precision in visual geolocalization at very large scales. Previous methods fall short due to the inherent tradeoff between accuracy and applicability. Classification methods must fall back to coarse partitions to train with current-day hardware. Retrieval methods suffer from unfeasibly large databases at inference time. We introduce a novel, hybrid approach that synergizes classification principles with cross-view retrieval by learning rich, ground-view feature prototypes in the same feature space as overhead feature embeddings. Our extensive evaluations on a continent-scale deployment across Western Europe demonstrate >68% top-1 recall accuracy within ~200 meters , establishing the feasibility of fine-grained geolocalization at an unprecedented scale.

## Acknowledgments

We thank Bernhard Zeisl for his feedback on early ideas and manuscript. We thank Matteo Balice for conducting experiments for the final version of the paper, including the impact of encoders and their initialization, the generalization capability to phone images, the robustness to temporal changes, and the comparison to language models.

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

# Appendix

We first discuss the societal impact of our work and later present additional ablation studies that motivate the design decisions of our approach, shedding light into its inner workings. We then evaluate the image embeddings learned by our model, study the data distribution of our dataset, and show extensive qualitative examples.

## A    Societal Impact

This work addresses the research question of geolocating images over very large geographical areas (countries, continents) without the need for rough priors, like GPS. A solution to this problem will undoubtedly raise concerns about privacy, surveillance, discrimination, and personal safety. While these concerns apply to any other works in the field, which we build and improve upon, they grow larger by scaling up the size of the database. As the potential of misuse is significant, we offer this as a proof-of-concept only, and will refrain from releasing model weights to the public. We note, however, that the same risks apply to any current visual place recognition (VPR) systems, which also typically perform best (albeit at a prohibitive cost).

Another potential misuse is in the training set, which covers a vast area of public places (streets, houses), including humans, animals and cars. Our data is anonymized, blurring faces and license plates in order to prevent leaking this information to the model.

On the other hand, we highlight the potential capabilities of such a system, which could enable novel applications in autonomous systems and augmented or virtual reality. It could also enable the creation of much larger 3D vision datasets by helping pose arbitrary images (in conjunction with more traditional solutions like Structure-from-Motion), a process that is currently very time-consuming and typically rejects a very large fraction of images. It also helps push the envelope on the understanding of geospatial patterns from multiple modalities (ground and aerial images). Finally, it offers very significant compute savings over retrieval-based systems (VPR), which are the state of the art in visual geolocalization.

## B    Additional Evaluations

### B.1    Ablations

**Ablation—Losses (Table 9).** We study the impact of the loss terms under different evaluation settings: (a) ground-to-aerial cross-view retrieval, (b) cell classification, and (c) our hybrid cell prototypes. All terms significantly contribute to the accuracy of our approach. Removing the edges between ground and aerial embeddings harms cross-view localization performance most because they get only indirectly constrained through the prototypes. The most important edge is between the ground images and cell prototypes, behaving as a global, spatial memory. However, this memory is limited by the actual density of samples in the cell, which acts as a bottleneck. Aligning ground images jointly to aerial embeddings and prototypes yields large improvements, especially on prototype retrieval. Empirically, we observed that this reduces overfitting between ground images and prototypes, which

Table 9: **Loss terms.** We study the impact of the loss components between ground-level (G), aerial (A) embeddings, and cell prototypes (P), on recall@$K$@200m on BEDENL, under different evaluation settings (a-c). We highlight the **best** and second best, per column. The bottom row is our final model.

| Terms | | | (a) Cross-view | | | (b) Prototypes | | | (c) Hybrid | | |
|:---:|:---:|:---:|:---:|:---:|:---:|:---:|:---:|:---:|:---:|:---:|:---:|
| G-A | G-P | A-P | $K$=1 | $K$=5 | $K$=100 | $K$=1 | $K$=5 | $K$=100 | $K$=1 | $K$=5 | $K$=100 |
| ✓ | | | 39.2 | 53.6 | 74.1 | N/A | N/A | N/A | 39.2 | 53.6 | 74.1 |
| | ✓ | | N/A | N/A | N/A | 47.2 | 59.3 | 74.5 | 47.2 | 59.3 | 74.5 |
| ✓ | ✓ | | 42.3 | 56.0 | 75.2 | 56.4 | 67.8 | 81.0 | 58.3 | 70.0 | 84.6 |
| ✓ | | ✓ | 46.4 | 59.8 | 78.6 | 40.7 | 54.6 | 73.9 | 47.0 | 60.4 | 79.6 |
| | ✓ | ✓ | 15.8 | 26.4 | 51.6 | 47.7 | 60.5 | 77.0 | 47.7 | 60.5 | 77.0 |
| ✓ | ✓ | ✓ | **49.7** | **63.3** | **81.0** | **57.1** | **68.6** | **81.8** | **60.3** | **71.6** | **85.6** |

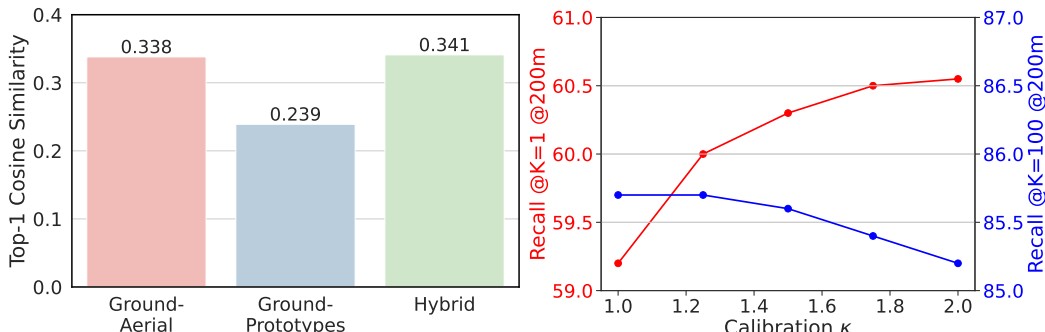

Figure 7: **Ablation of the calibration factor. Left:** Average top-1 cosine similarities. **Right:** Impact of the calibration factor $\kappa$ on top-1 (red) and top-100 (blue) recall at 200m.

Table 10: **Generalization to gaps in the map.** We expand the database with aerial-only embeddings on cells not covered by ground-level images. We report recall both (a) in areas where training data, and thus cell prototypes, are available, and (b) where cell prototypes are not available, a common failure case of approaches that rely only on ground-level images, which we bridge via aerial embeddings. Our hybrid method is able to generalize well to unseen areas in the map.

| Method | (a) BEDENL | | | (b) BEDENL gaps | | |
|---|---|---|---|---|---|---|
| | Recall@$K$@200m | | | Recall@$K$@200m | | |
| | $K$=1 | $K$=5 | $K$=100 | $K$=1 | $K$=5 | $K$=100 |
| SALAD [24]-Aerial | 34.2 | 47.1 | 67.2 | 25.8 | 36.9 | 57.0 |
| Ours (full) | 55.1 | 67.5 | 83.1 | 35.6 | 46.9 | 64.9 |

is more common in cells where the data is sparser. The aerial embeddings smooth the feature space and thus reduce the dependency on sampling density, countering overfitting.

If the edge between aerial embeddings and cell prototypes is missing, this introduces an asymmetry whereby global constraints come only from the ground-level embeddings, significantly harming performance in cross-view retrieval. The model benefits from regularized aerial embeddings which better constrain the space, serving as a proxy for hard negative mining between ground and aerial images. Combining all loss terms strikes a strong balance between cross-view and prototype retrieval performance. One downside of the full model is its requirement to compute the full similarity matrix to all cell prototypes twice (once for the ground images and once for aerial) during training.

**Ablation—Calibration prototypes and aerial embeddings (Figure 7).** One important hyperparameter in our study is the calibration factor we use when combining the aerial embeddings with the prototypes, *i.e.*, for our 'hybrid' model, at inference time. One key insight here is that the actual similarity scales are different: queries show about 1.5 times larger similarity to the aerial embeddings, both on the training and test sets. The left panel in Fig. 7 illustrates this observation. We partially attribute this to the different granularity between aerials (L16) and cell prototypes (L15), as the coarser granularity of the prototypes means that they need to average over larger areas and thus more visual content, yielding lower similarity scores to each query. On the right panel we show recall metrics for different values of the calibration factor $\kappa$—recall@200m for both the top-1 and top-100 candidates. The best trade-off in recall is observed at approximately $\kappa = 1.5$, which corresponds to the offset factor between the similarities. This supports our design choice to select $\kappa$ based on this delta between the two similarities. Overall, recall performance demonstrates robustness to changes in the calibration factor.

**Ablation—Missing training data (Table 10).** A benefit of aerial embeddings over prototypes is their ability to generalize to unseen cells. In Table 3 (in the main paper) we discussed how this helps the model generalize to different countries. In practice, we are more interested in the generalization capabilities to *gaps* in-between prototypes, *i.e.*, smaller regions or 'holes' in our database where we might have aerial coverage but not enough ground-level images to build cell prototypes.

In the main paper, we evaluate only on cells where we can train the model, omitting cells for which we have enough *test* data (from 2023) but not enough *training* data (from other years) — this allows us to provide a fair comparison between cross-view and prototype-based retrieval, as they use the

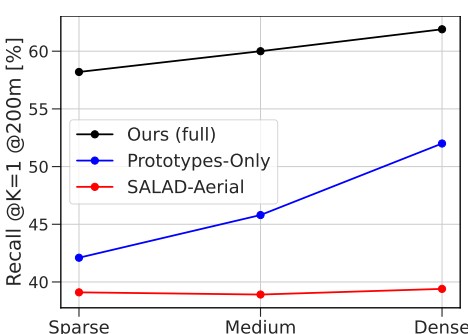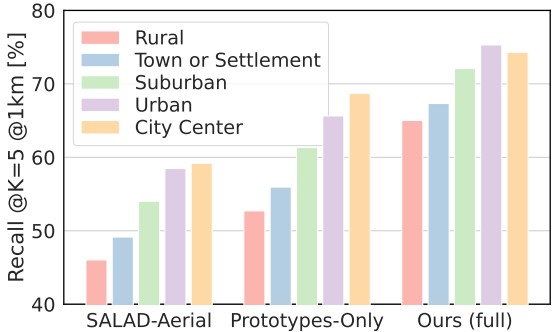

Figure 8: **Factors impacting the accuracy. Left – density of StreetView training images on** BEDENL**:** In areas with few images, prototypes focus on irrelevant details and thus exhibit a lower accuracy. They are are however more robust when sufficient data is available (*e.g.*, city centers). Ground-aerial retrieval is not affected by this factor. Our hybrid approach combines these benefits and delivers the largest improvements in areas with little training data. **Right – population density on** EuropeWest**:** For all methods, the performance is higher in more densely populated areas (city centers), which generally have more distinctive visual information than country roads or highways. Our approach delivers the largest improvements in rural areas.

same subset of the data. In this experiment we aim to increase the test coverage *beyond* that of the training set. We collect 166k test images that are at least $200m$ from their closest prototype center in BEDENL, and from the year reserved for the test set (2023). We extend the aerial database to contain these areas, increasing the database size from $4.8M$ to $8.5M$ $L$=16 cells, and evaluate our best model against SALAD [24]-Aerial. The results in Table 10 show that our method can localize images in these areas, although with a significant performance drop. This captures the use-case where we are missing cell prototypes and must fall back to pure cross-view retrieval. Note that compared to the results in Table 1 (in the main paper), the in-domain performance (*i.e.*, for cells with a prototype) also drops slightly because of *almost doubling* the size of the database.

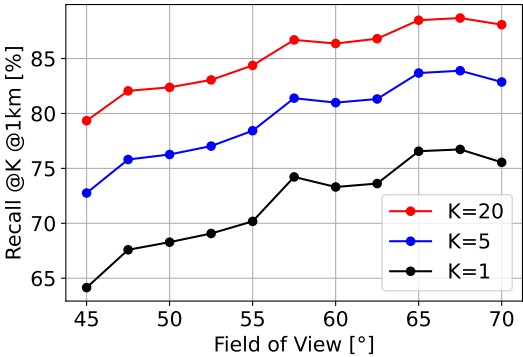

Figure 9: **Impact of the field-of-view on the accuracy on** BEDENL**.** A larger field of view results in a higher accuracy.

Table 11: **Impact of temporal changes.** We evaluate queries of BEDENL that are captured in years 2023 and 2025 and in locations with data available in both years (43.8k S2 cells). The models are trained on data from 2017-2022 & 2024. The performance is similar for both sets of queries, showing that our model is robust to temporal changes.

| Year | Method | Recall@$K$@200m | | |
|------|--------|------|------|------|
| | | $K$=1 | $K$=5 | $K$=10 |
| 2023 | Ours (full) | 60.3 | 71.6 | 85.6 |
| | Ours (DINOv3-L) | 67.9 | 77.2 | 87.9 |
| 2025 | Ours (DINOv3-L) | 67.3 | 76.7 | 87.8 |

**Impact of the density of training data (Figure 8-left).** We group queries by the number of StreetView training images available in their corresponding cells, from 'sparse' to 'dense', and report the recall@1@200m. Our hybrid approach is the most robust to this factor because it is able to combine both overhead and ground-level cues.

**Impact of the population density (Figure 9-right).** We group queries by the population density of the areas in which they are located, from lowest (rural roads or highways) to highest (city centers) density and report the recall@5@1km. Our hybrid approach is the most robust to this factor.

**Impact of the field of view (Figure 9).** We group queries by their field of view, which is randomly sampled in $[45°, 75°]$ and report their recall@1km. We notice that the localization accuracy generally increases with the field of view, as more visual context helps to disambiguate the location.

**Impact of temporal changes (Table 11).** We have trained our models with images that have been captured in years 2017-2022 & 2024 such that queries of the test set were captured in year 2023. To

Table 12: **Cross-Area Visual Place Recognition in** `Portugal`. We perform an additional experiment that compares visual place recognition between ground view-images to cross view retrieval, both with a large spatial domain gap, in a country not covered by the training set, `Portugal`. Visual place recognition generalizes significantly better. Our model outperforms the popular VPR baseline SALAD [24], also trained on StreetView imagery.

| Evaluation | Method | Training | Portugal Recall@$K$@200m | | | dim. |
|---|---|---|---|---|---|---|
| | | | $K$=1 | $K$=5 | $K$=100 | |
| ground retrieval | SALAD – official weights [24] | `GSV-Cities` | 27.3 | 36.2 | 53.9 | 8448 |
| | Ours (prototypes-only) | `BEDENL` | 47.6 | 58.7 | 74.8 | 2176 |
| | Ours (full) | `BEDENL` | 50.3 | 62.2 | 78.8 | 2176 |
| aerial retrieval | SALAD [24]-Aerial | `BEDENL` | 7.4 | 13.5 | 32.8 | 2176 |
| hybrid | Ours (full) | `BEDENL` | 10.8 | 18.5 | 39.6 | 2176 |

evaluate the impact of temporal changes, and to reflect a more practical use case in which queries are captured only after the database, we now evaluate our model on additional test images captured in year 2025. To mitigate any spatial bias, we only consider here the subset of queries that were captured in the same S2 cells in 2023 and 2025. The results, reported in Table 11, show that our model localizes images from both years equally well and is thus robust to temporal changes.

**Missing SALAD evaluation on** `BEDENL`. In Table 1, we did not report numbers for ground retrieval using SALAD [24] with official weights and $D = 8448$ because of the excessive database size (>5TB). For completeness, we reran this experiment using more resources, achieving localization recall within 200m of 19.0% / 25.3% / 39.6% for K = 1 / 5 / 100.

## B.2 Cross-Area Visual Place Recognition

**Setup:** We perform an additional study on classic image retrieval. We evaluate our model trained on `BEDENL` in a country not included in our training set, `Portugal`. This dataset consists of 18.8M images spaced 40 meters apart, similar to the distribution of the `BEDENL` training split. We evaluate on 197k test images from a different year. This benchmark evaluates the strength of learned image embeddings to large viewpoint and seasonal changes. **Baselines:** Unlike in the paper, we here use the *official weights of SALAD* [24]. This model is trained on the smaller Street-View dataset GSV-Cities [84], which contains images from major metros around the globe (including Lisbon, which is part of this test set). We further add our own cross-view retrieval baselines to this benchmark (database size 1.2M aerial images). Note that the features produced by SALAD are 4x larger than ours — too large in fact to run over `UK+IE`, which we used in the main paper. **Results:** We report the benchmark results in Table 12. Notably our learned embeddings outperform SALAD [81] trained on GSV-Cities [84]. This can be explained by the extensive amount of rural images in the test set, a domain not covered by GSV-cities [84], which pose a major challenge in country-wide geo-localization. The image embeddings learned from prototypes only generalize equally well to new domains, which is in contrast to the evaluation in-domain (*i.e.*, on `BEDENL`). Overall, image-based retrieval, despite the large viewpoint changes, still generalizes much better than cross-view retrieval to new areas. Notably, the gap between VPR and cross-view retrieval is significantly larger than in training areas (*i.e.*, for `BEDENL`, Table 1), as the model has to overcome both spatial, temporal, and viewpoint domain gaps.

## B.3 Fine-grained Localization results

We provide an additional table that reports localization results at the finer 100 m threshold in Table 13. Note that our prototypes alone are too coarse for an evaluation at this threshold, usually spanning a region of $200 \times 200$m at $L$=15 (for compute reasons). The aerial embeddings, in contrast, are at a finer threshold ($L$=16), *i.e.*, a quarter of the area covered by the cell prototypes, thus enabling finer-grained localization — given the space limits we reported only results at 200 m in the main paper, which allows us to make direct comparisons for all variants.

Table 13: **Fine-grained localization.** We report recall for our best model on the two main datasets used in the paper, BEDENL and EuropeWest, at a *finer 100 m threshold* rather than 200 m, which closely aligned to the finer-grained cells ($L$=16) used for cross-view retrieval. Notably, our prototypes are coarser at $L$=15, with an approximate size of $200 \times 200$m. Despite this, combining coarse prototypes with finer aerial embeddings greatly boosts recall even at finer thresholds.

| Ours (full) | Level | BEDENL Recall@$K$@100m | | | EuropeWest Recall@$K$@100m | | |
|---|---|---|---|---|---|---|---|
| | | $K$=1 | $K$=5 | $K$=100 | $K$=1 | $K$=5 | $K$=100 |
| Cross-view retrieval | L16 | 45.5 | 60.1 | 78.6 | 40.3 | 55.8 | 75.3 |
| Prototype retrieval | L15 | 24.4 | 29.6 | 34.6 | 23.4 | 30.2 | 36.3 |
| Hybrid | L16 | **54.3** | **69.6** | **84.4** | **50.2** | **67.5** | **83.0** |
| Hybrid (DINOv3-L) | L16 | 60.6 | 77.4 | 89.2 | 59.2 | 76.9 | 87.9 |

Figure 10: **Localization errors for queries in** BEDENL. Left: SALAD [24]-Aerial, Middle: Ours (full), Right: Ours (Prototypes). Our method mostly improves especially in rural areas, where ground-level training data is sparser.

Our proposed hybrid evaluation that averages cell prototypes with aerial embeddings yields significant improvements. Note that we bridge the granularity gap by nearest-neighbor interpolation, *i.e.*, four $L$=16 aerial embeddings are paired with the same $L$=15 cell prototype.

## B.4 Spatial error distribution

To better understand the improvements of our hybrid retrieval method, we illustrate the localization errors spatially. We conduct this experiment for the cross-view retrieval baseline SALAD [24]-Aerial, our best baseline that does not use aerial images (Ours (prototypes-only)), and our full model. The results are illustrated in Fig. 10. Notably our full model (middle) achieves improvements uniformly in all areas, which are mostly rural cells with low data density. There, the aerials, which are almost unaffected by data density, yield large improvements over cell prototypes, which needs to remember content seen during training. Prototypes, on the other hand, are inherently globally discriminative, and exhibit strong performance in more densely sampled areas of our datasets, which is weakly correlated with population density.

## B.5 Comparison to Vision Language Models

There has been increased interest in using Vision Language Models to solve the geolocalization problem. As such, we also compare our approach to a state-of-the-art model, Gemini 2.5 Pro [85], using the following prompt, derived from GeoBench [86]:

*You are participating in a geolocation challenge. Based on the provided image: 1. Carefully analyze the image for clues about its location (architecture, signage, vegetation, terrain, etc.) 2. Think step-by-step about what country this is likely to be in and why 3. Estimate the approximate latitude and longitude based on your analysis Hint: the image is located in one of the following Western European countries: Spain, Portugal, France, Belgium, Netherlands, Germany, Czechia, Austria,*

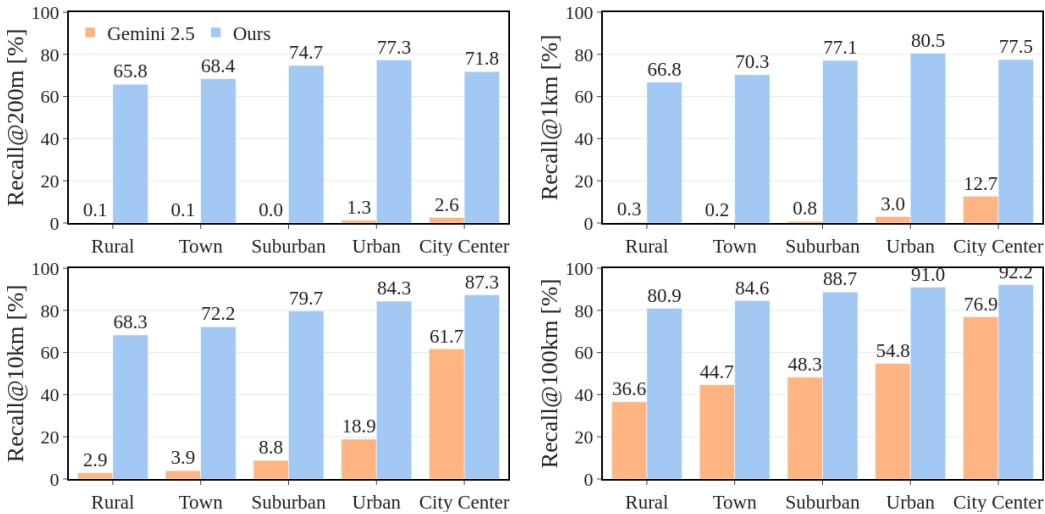

Figure 11: **Comparison to Gemini 2.5 on** `EuropeWest`. We conduct an additional experiment, and compare our best model (DINOv3-L) with Gemini 2.5 Pro [85]. We report the localization accuracy within 200m, 1km, 10km and 100km, for queries associated with different levels of population density.

*Switzerland, Italy. You do not have access to internet nor StreetView so do not hallucinate a reverse image search or StreetView lookup. Take your time to reason through the evidence. Add in-depth reasoning 'explanation' field, clearly explaining why you chose this location instead of others.*

In Figure 11, we compare the localization accuracy on a subset of queries from `EuropeWest` to our best model (DINOv3-L backbone). Both on rural and urban queries, our method is able to accurately localize queries up to 200m, while Gemini is generally only able to provide coarser estimates.

## C Implementation and baselines

**Performance optimization.** Training our full model requires computing the similarity between both ground-view and aerial embeddings to all the prototypes. The performance bottleneck is two-fold: First the actual dot product, and second the all-to-all transform to gather the sharded similarities to the correct device. Improving inference speed on the actual similarity computation would require heuristics (e.g. training and maintaining a shortlist [87]), which would drastically increase the complexity of our method, and we thus refrain from doing so. Experimentally we found the all-to-all transform to be the actual bottleneck in our system, as it involves transferring the full similarity vector per batch element, twice. We alleviate this by first broadcasting and replicating all modality embeddings to each device, and then compute the similarity to the shard of prototypes on the specific device. We then compute the loss directly on the device, which improves training speed from $0.7$ steps/sec to $1.0$ steps/sec, without any impact on accuracy. However, this still requires maintaining the full gradient to all prototypes, which is both inefficient and harms accuracy, as many prototypes not visible in the batch get tiny, noisy updates. Furthermore, one can utilize approximate nearest-neighbor search within each device to mine hard negatives, and only add these elements to the loss. This improves the training speed to $1.2$ steps/sec on `BEDENL`, while also achieving slightly higher accuracy ($+0.9$ top-1 recall at 200m). However, for simplicity we report all results without approximate nearest neighbor search in the paper. For reference, our prototype-only baseline runs at $1.7$ steps/sec, and our cross-view retrieval baseline (SALAD [24]-Aerial) at $1.5$ steps/sec. On our hardware (128 TPUs), localizing a query image (encoding + retrieval) on `EuropeWest` takes around 0.4 sec.

**Baselines.** We discuss the (re-)implementations of the major baselines that we compare against:

- **Fervers *et al*. [30]:** We adopt the same multi-head attention head, and the decoupled, bidirectional InfoNCE loss with label smoothing factor $0.1$. For a fair comparison with our method, and in contrast to the original paper, we replace the ConvNext [88] backbone with a ViT [46], similar to all other baselines. We use a temperature $\tau = \frac{1}{36}$ as in the original paper. The original paper used a spacing of $5m$ between training images, which proved infeasible

for us to run at this scale. We therefore equalize the data for a fair comparison. Similarly, the original paper adopted a cell size of 30x30 meters, which would increase size of the database by a factor of 10. We thus evaluate their method on the same resolution as ours (100x100 meter), and adopt the offset accordingly. One core insight of Fervers *et al*. [30] is that an image pyramid per cell yields substantial improvements. However, when experimenting we found this to be a major performance bottleneck, reducing throughput from 1.5 steps/sec to $< 0.5$ steps/sec. Furthermore, this insight is orthogonal to our method and would improve every approach that relies on aerial images. We therefore use a single level, *i.e*., images of $256 \times 256$ px and a resolution of $0.6\frac{m}{px}$, for both the baseline and for our model. We train the network for an equal amount of steps as our method. The authors also propose a look-ahead hard example mining (HEM) strategy, yet the authors note that this is not required with large batch sizes (our setup uses a batch size of 8192), and it is a performance bottleneck which requires an additional forward pass per batch. Instead, we try to strengthen the baseline by performing an offline hard-negative mining (on a trained model) using the aerial embeddings. For each cell, we encode its aerial image and find the top-k most similar features from other cells. During training, we then load per element images from 64 of its neighboring cells, and run training for 2 full epochs.

- **SALAD [24]-Aerial:** This baseline uses the same architecture as the original SALAD [24], both for the aerial and ground-level encoder (weights are not shared). In contrast to the original model, we initialize from iBOT [42] weights and finetune the entire network to account for the large domain gap between ground and aerial images. Similar to our work, we adopt the Multi-Similarity Loss [34], but without online negative mining. Instead, we contrast to all other elements in the batch. Similar to Fervers *et al*. [8], we use a bidirectional loss to contrast ground-level to aerial images and vice-versa. The remaining hyperparameters are identical to our full implementation.

- **Haversine loss [1]:** We adopt the haversine loss from PIGEON [1], which is a form of spatial label smoothing. We change the haversine temperature from $\tau = 75$ km in the original paper, tuned for coarser localization, to $\tau = 200$ m, which we empirically found to provide a nice trade-off between robustness and accuracy. The architecture and head are identical to our network, and we use l2-normalized embeddings with a learned temperature initialized to $\tau = 0.01$.

- **Hierarchical loss [2]:** OSV-5M [2] has demonstrated that a simple hierarchical loss on a quad-tree yields substantial improvements. We adopt this baseline in our evaluation, and adapt it with a tree of height $h = 4$ and using every second level in the loss, *i.e*., we supervise the sum of probabilities at $L$=15, $L$=13, $L$=11, and $L$=9, and use a learned temperature initialized to $\tau = 0.01$. We use the same architecture and training setup as our main method.

**Critical hyperparameters.** We found that the most critical hyperparameters besides the learning rate are in the loss function. Of these, the base parameter $\lambda = 0.2$ has the largest impact on performance, controlling the push on the similarities. The parameters $\beta = 100$ and $\alpha = 2$ control the balancing between positive and negative examples. During training, the network tends to first push the prototypes apart to be close to orthogonal, and then increase the similarity to the respective ground-level and aerial embeddings in their region.

**Efficient experimentation.** As training these networks from scratch is expensive, we found that pre-training networks for cross-view retrieval, and then fine-tuning the network with prototypes yields equal performance at a fraction of the time. There, we initialize the backbones from the pretrained weights, and randomly initialize the weights of the heads and prototypes. In these experiments, we also found it beneficial to increase the learning rate of the head to 0.01, similarly as for the cell codes.

## D   Datasets

**Comparison to public datasets.** We provide a qualitative comparison to public datasets in Table 14. Notably, no existing dataset has sufficient spatial extent and density for fine-grained, continental-scale geo-localization. The dataset introduced by Fervers *et al*. [30] is most similar to ours, yet their actual spatial distribution of training and test images is less dense and biased by the viewpoint (the authors acknowledge that the "frontal street-view perspective is heavily overrepresented" in their dataset [30]). Instead, we rely on random crops from $360^o$ panoramas to model arbitrary viewpoints.

Table 14: **Comparison with existing public datasets.** They generally exhibit neither sufficiently dense coverage nor the spatial extent as large as our dataset. Some datasets do not include aerial imagery or only forward-facing ground-level imagery.

| dataset | training | | | evaluation | | | aerial images | ground images |
|---|---|---|---|---|---|---|---|---|
| | images | spacing | countries | images | split | size | | |
| EuropeWest | 470M | 40m | 10 | 4.5M | temporal | continent | ✓ | random |
| Fervers *et al.* [8] | 72M | 5m | 2 | 11M | cross-area | state | ✓ | forward-facing |
| OSV-5M [2] | 5M | 1km | 225 | 210K | spatial | global | – | forward-facing |

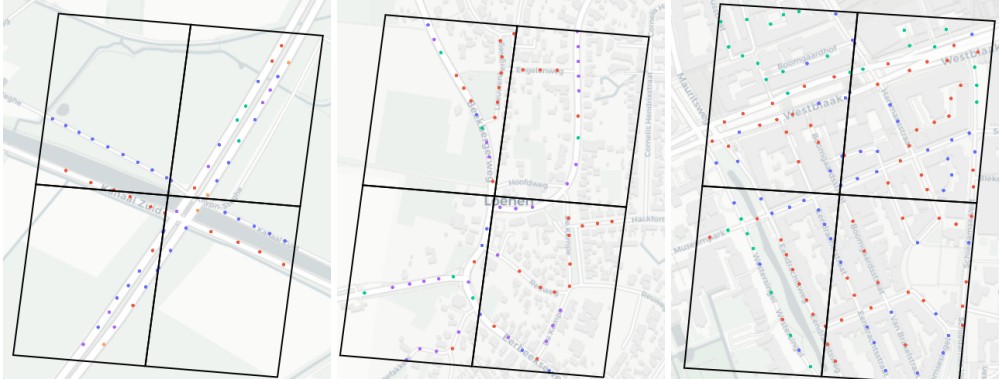

Figure 12: **Sampling of ground-level training images.** We illustrate the sampling within 4 $L$=15 cells for 3 examples. Each dot represents a panorama and each color corresponds to a different year in 2017–2024. We sample panoramas such that they are at least 40 m apart, which is a good trade-off between density and coverage. Left: intersection in a rural area, Middle: rural town, Right: a major city in BEDENL.

**Pinhole rendering.** We render $224\times224$px pinhole images from stitched panoramas with height 768px (thus minimizing aliasing). To gain robustness to different intrinsics and viewpoints, we randomly sample a roll in range $[-10°, 10°]$, a pitch in $[-5°, 15°]$, and a field of view in range $[45°, 75°]$. For the yaw, we sample a random offset per panorama, stratify the yaw, (for example, four yaws at 90°increments), and randomly perturb each yaw with a uniform random offset.

**Local sampling.** As discussed in a previous section, we employ spatio-temporal farthest point sampling to maximize coverage in both axes. We illustrate our sampling in Fig. 12. Our approach avoids oversampling cells that are only crossed by a few streets, while yielding dense coverage in metros. This strikes a nice balance between accuracy and efficiency, thereby enabling the large scale experiments conducted in this work. However, we would like to point out that this inherent adaptive density might still not be sufficient in metros, as it 1) does not account for increased occlusions (from traffic and denser settlements) and more frequent temporal changes (e.g. construction sites) in urban areas, and 2) is only weakly proportional to the actual population density. This motivated us to perform additional experiments on BEDENL+, a dataset that mixes additional urban training samples to the existing dataset, BEDENL (BEDENL+ is a superset of BEDENL).

**Spatial coverage and density.** We illustrate the spatial and temporal coverage in Fig. 13. Most cells in our dataset have low data variety, averaging at around 20 panos per cell. The density is significantly higher in urban and suburban areas. Contrary to the density in panoramas, the amount of unique runs these panoramas were captured in shows a significantly different behaviour: We have the largest temporal diversity on highways. This also supports our qualitative observation of increased accuracy on highways (see e.g. Fig. 5), which at first sight is counter-intuitive because of the lack if visual landmarks there.

**Holes in dataset.** Our dataset exhibits a few rectangular holes in the dataset. In these areas, our dataset lacks aerial image coverage from the same sensor, and we thus exclude this from training. During evaluation, however, as can be seen in Fig. 5, we utilize satellite imagery of lower quality there. While this does impact the accuracy, our method is still able to correctly localize a substantial part of images there. We therefore suspect that mixing aerial and satellite imagery would further increase the robustness of our localizations system.

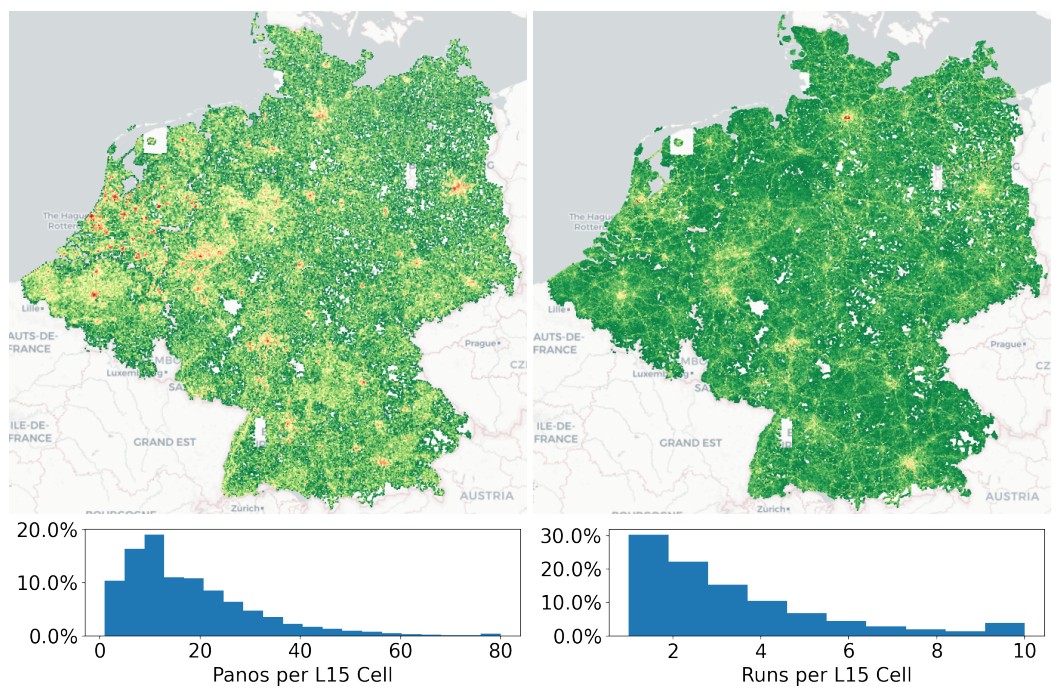

Figure 13: **Data density.** The training data densely covers BEDENL. The density of panoramas (left column) is weakly proportional the the population density, while temporal diversity from runs (right) is largest in urban centers and on highways. On average, a cell in our dataset has 20 panos (4 sampled views per pano) from 3 unique runs. Both have a direct impact on the final localization accuracy, as shown by Fig. 4.

Table 15: **Number of training and evaluation images per country (millions).**

|          | PT   | ES   | IT   | AT   | CH  | DE    | FR    | BE   | NL   | CZ   |
|----------|------|------|------|------|-----|-------|-------|------|------|------|
| Training | 19.0 | 52.3 | 76.9 | 16.4 | 8.3 | 115.5 | 149.5 | 13.5 | 17.0 | 14.5 |
| Test     | 0.2  | 0.5  | 0.7  | 0.2  | 0.1 | 1.2   | 1.4   | 0.2  | 0.2  | 0.1  |

**Statistics by country.** In Table 15 we provide detailed statistics about the number of train and test images per country. Notably, our test set is sampled uniformly over space. While it could be argued that this biases results towards sparse, rural areas, which usually have harder queries, we believe this accurately reflects how one would assess a true global localization system, which should exhibit spatially uniform performance.

## E  Visualizations

We qualitatively show query examples in the EuropeWest dataset, labelled by their top-1 localization error (in km), see Fig. 14. We compare retrieval to the strongest cross-view (G-A) and classification-only (G-P) baselines on this dataset. Notable, cross-view retrieval methods achieve higher accuracy in rural areas with few bird's-eye occlusion obstacles, while cell prototypes excel in urban areas with large, vertical facades. Our method combines the best of both worlds by relying on the factor that best describes the respective area.

The last couple of rows show failure cases. Notably, low-texture regions such as the border wall of highways or zoomed-in images, large dynamic occlusions (trucks and cars), and repetitive vegetation are limitations of our approach. Overall, we qualitatively made the observation that the network tends to localize images very well if 1) the road is visible and ahead, which coincides with 2) the image has large depth, and therefore can observe distant features. We believe that these distant landmarks are beneficial as they are easier to observe from different angles, and can help to coarsely remember locations, especially in rural areas.

Fig. 15 shows the top-5 retrieved aerial images of our method. Notably, the network is able to utilize patterns in the vegetation, such as the spacing between trees (row 3), or features at large depth (row 5) to robustly find the correct area. Failure cases are typically close-up images (third-last row), or ambiguous queries such as empty highways, or very large occlusions.

We finally analyze the PCA visualizations of our network and one major ablation in Fig. 16. Supervising just prototypes yields smooth and very informative prototypes that clearly encode semantic and geological entities. Empirically, we observe PCA features to be more informative the coarser the cells are (*i.e.*, the stronger the information bottleneck).

Finally, we show self-similarity patterns between cell prototypes in Fig. 17. While the prototypes are almost fully orthogonal to each other, which helps localization, they are locally similar, hinting that the network indeed uses coarser geospatial patterns for localization.

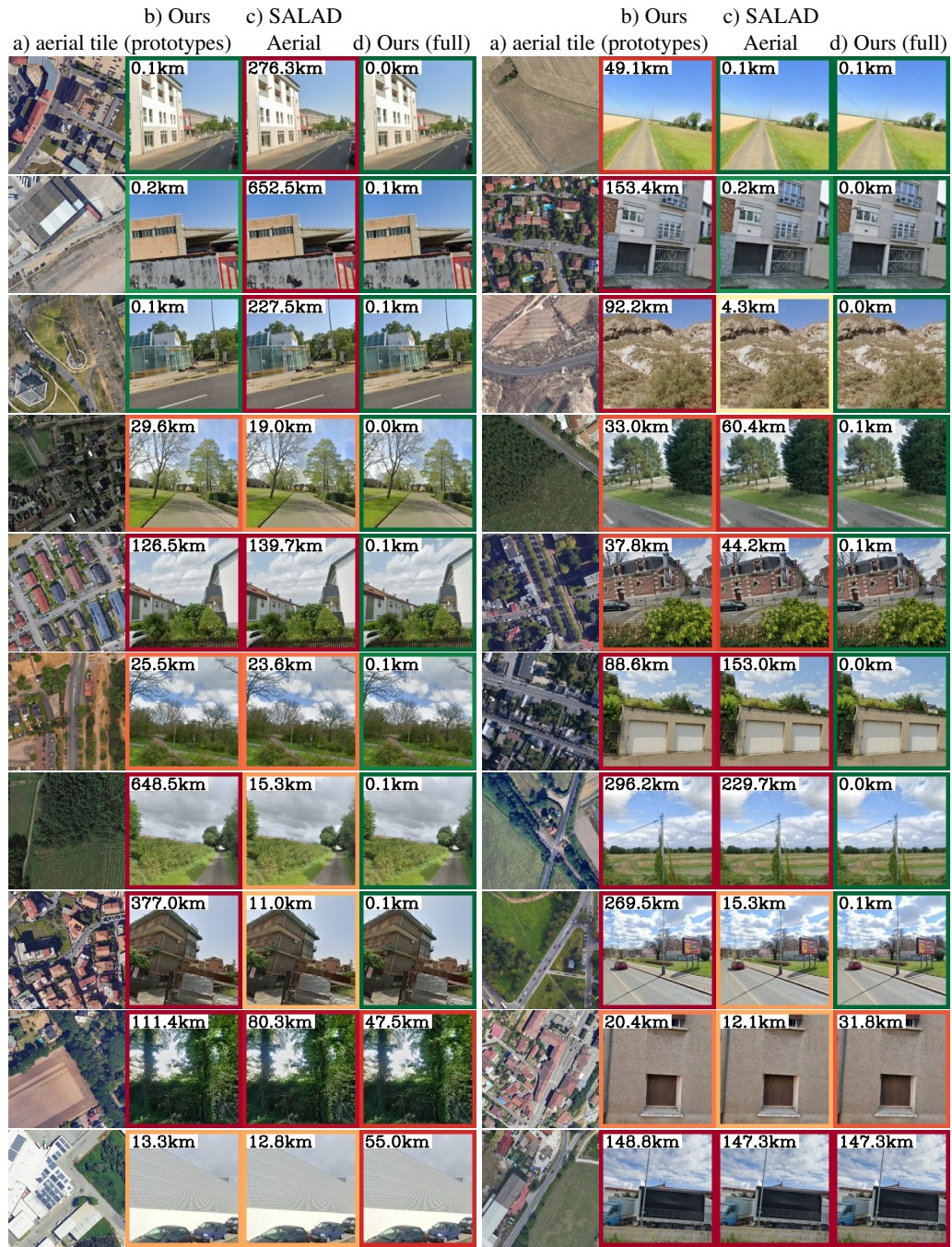

Figure 14: **Localization errors of queries in** `EuropeWest.` a) Database aerial image for the ground truth location, b) Query image and rank-1 error for our prototype-only variant, c) Query image and rank-1 error for the baseline SALAD [24]-Aerial, d) Query image and rank-1 error for our full hybrid approach. Our approach is able to correctly localize ambiguous rural cells by combining ground- and aerial cues, and is robust in many scenarios. Failure occurs in scenarios with occlusion by transient objects (cars, trucks) and queries with narrow field of view. In general, the method performs significantly better the less the view is obscured.

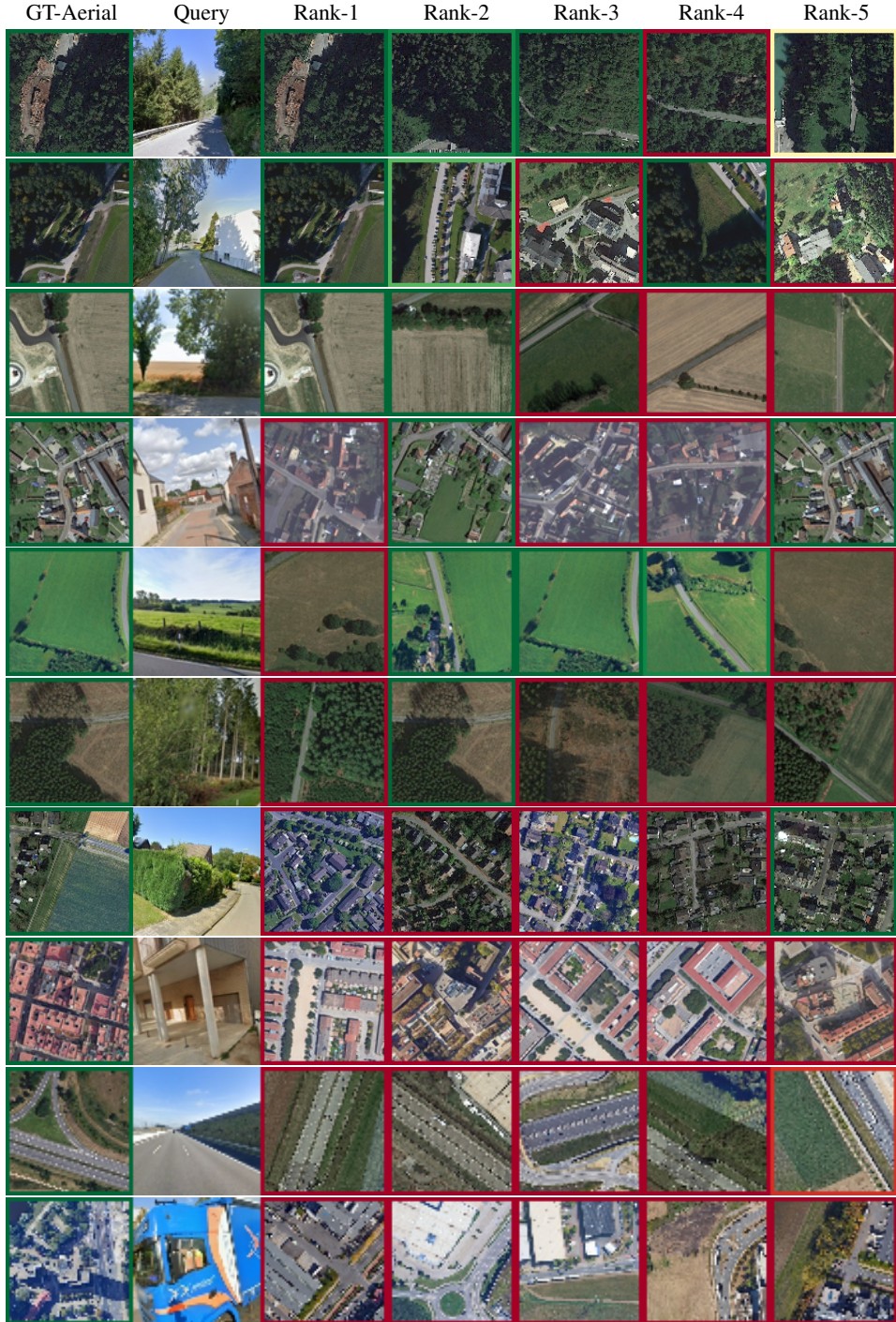

| GT-Aerial | Query | Rank-1 | Rank-2 | Rank-3 | Rank-4 | Rank-5 |

Figure 15: **Cross-view retrieval results in** `EuropeWest`**.** Starting from the left, we show an aerial tile of the correct cell, the respective query image to be localized, and the aerial tiles corresponding to the top-5 cells predicted by our full model. The image frames are colored following Fig. 10—green is < 100 m. Our approach is able to localize images both in rural and urban settings and typically yields multiple close-by retrievals in the top-k predictions, thanks to our frustum and cell interpolations. Common failure cases are close-up images, vegetation and large occluders (cars, trucks).

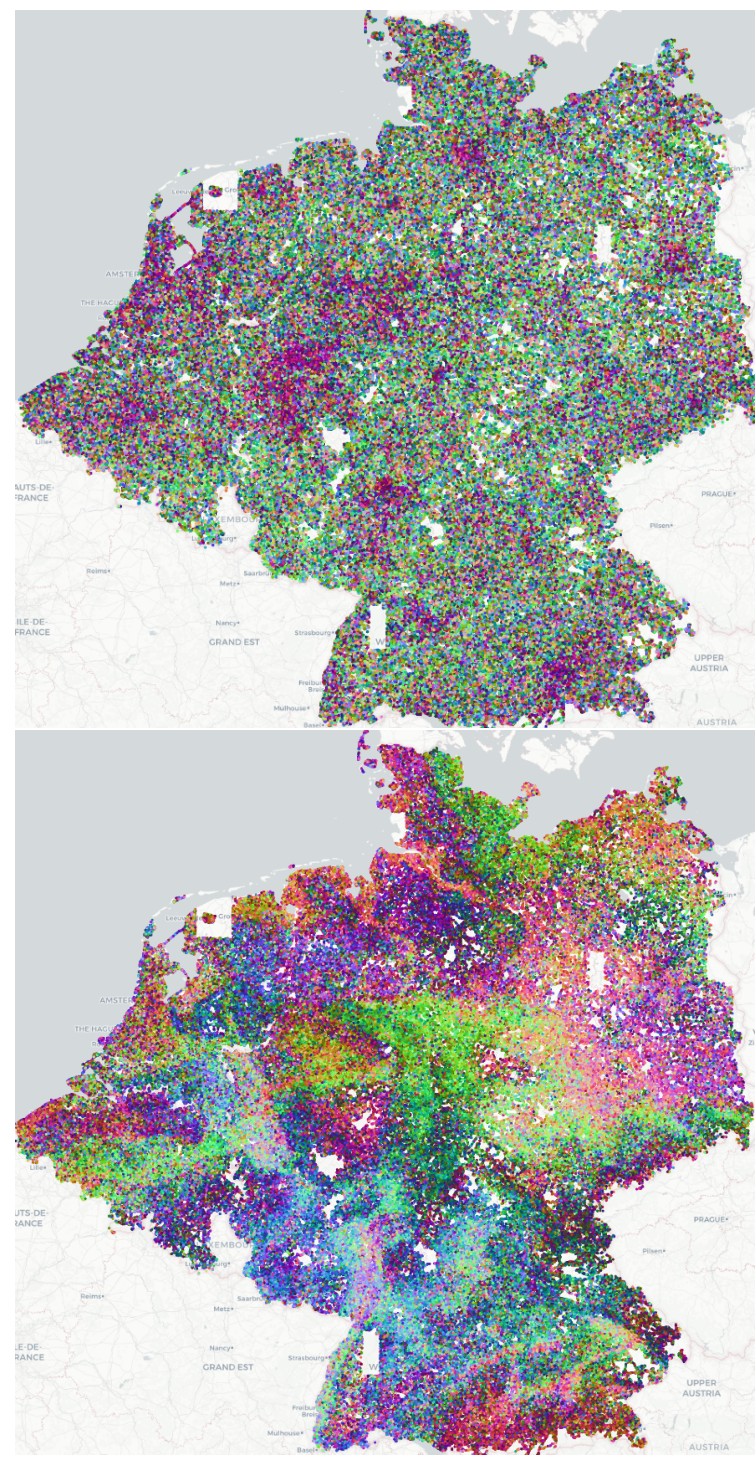

Figure 16: **PCA visualizations of the cell codes over** BEDENL. We compare the prototypes from our full model (top) to the prototypes from the prototype-only variant (bottom). While the prototypes of our full model encode a lot of high-frequency information that boost the accuracy, only supervising the prototypes yields smoother and more informative features. It clearly encodes the river delta (Elbe) to Hamburg (top of the image) and can separates geological entities which are hard to observe in single images, such as the Black Forest in the lower left or the Alps at the very south.

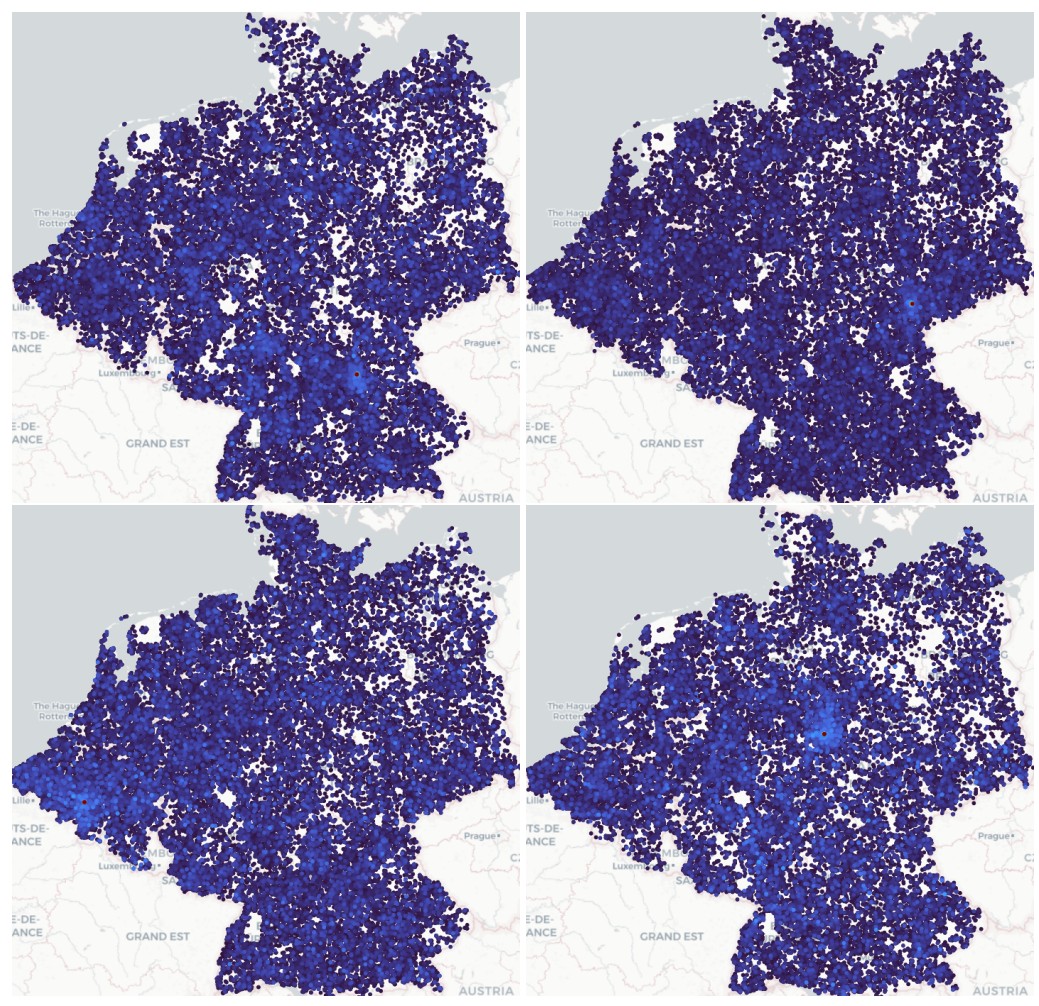

Figure 17: **Self-similarities between prototypes in** BEDENL. We show the self-similarities of 4 prototypes (red dots) to their top 50k neighbors. Red and blue correspond to a high and low similarities, respectively. The prototypes are almost fully orthogonal, yet locally smooth.

