# OpenReview forum: "Scaling Image Geo-Localization to Continent Level"
_NeurIPS.cc/2025/Conference — NeurIPS 2025 poster_

### Official Review · Reviewer_n8dV · 2025-06-26

**Clarity:** 4
**Significance:** 3
**Originality:** 3
**Rating:** 4
**Confidence:** 4

**Summary:**

This paper addresses the challenge of visual geo-localization: determining the precise geographic location of a query image at continent-wide scale, moving beyond the city- or region-level localization. The authors identify the core trade-off in existing methods: classification-based approaches are scalable but yield coarse, imprecise results (often >10km error), while retrieval-based methods are precise but computationally infeasible at a continent scale due to massive database sizes and the domain gap between ground and aerial views.
The key contribution is a hybrid approach that leverages the strengths of both paradigms. The method involves two main components:
1. Learned Cell Prototypes: The world is partitioned into S2 cells. A model is trained on ground-level images (like Google StreetView) using a proxy classification task to learn “prototypes”, rich feature embeddings that summarize the visual appearance of a geographic area.
2. Cross-View Fusion: These ground-level prototypes are fused with aerial image embeddings from the corresponding overhead tiles, producing compact and discriminative “cell codes” that incorporate both street-level and aerial context.

This fusion creates a compact and robust “cell code” for each location. At inference time, a query image is encoded, and retrieval is performed against this highly efficient database of cell codes. The authors conduct extensive experiments on massive datasets covering most of Western Europe (up to 470M images), demonstrating that their method achieves over 50% top-1 recall accuracy within 100 meters. This establishes a new state-of-the-art for fine-grained geo-localization at a continent scale, using a database that is 30-60x smaller than traditional retrieval methods.

**Questions:**

1) Could the authors provide a learning curve showing how the final model performance (e.g., Recall@1@100m) varies with the size of the training dataset? For example, by training on subsets of the EuropeWest dataset (e.g., 10%, 25%, 50%). This would provide crucial insight into whether the impressive results are entirely dependent on massive data or if the method provides significant benefits even at a smaller scale.

2)The paper acknowledges reduced performance in rural areas or under occlusion but offers limited insight into why specific images fail and whether these failures are recoverable. i) Are failures more due to visual ambiguity, lack of data, or aerial-ground mismatch? ii) Could confidence estimation or uncertainty modeling help reject low-quality predictions?

3) Could the authors clarify whether the “Prototypes-Only” model was trained independently, or if its results were obtained by omitting aerial features at inference time from the full hybrid model? If the latter, it would suggest that joint training with aerial supervision improves prototype quality, even when not used at test time, which would be an interesting insight worth highlighting more directly.

4) The paper demonstrates strong results over a multi-country region (10 countries in Western Europe), but it stops short of true continent-wide or global coverage. Could the authors comment on the feasibility of scaling the method to an entire continent (e.g., all of Europe) or even globally? Given the current reliance on 128 TPUv2 for training, do the authors envision ways to reduce computational costs that could make the approach more broadly deployable?

**Ethical Concerns:**

["NO or VERY MINOR ethics concerns only"]

**Final Justification:**

N/A

**Limitations:**

Yes

**Quality:**

3

**Strengths And Weaknesses:**

S1 - The use of a massive dataset covering a substantial part of Western Europe (470M images) provides a robust testbed that exceeds prior work.  The paper is supported by a strong supplementary material that includes comprehensive ablations on loss components (Table 7), the calibration factor (Fig. 6), and data density (Fig. 7). The authors convincingly demonstrate the quality of their learned representations by testing on completely unseen geographic areas (Portugal, UK+IE) and a different data domain (pedestrian-view Trekker dataset), showing strong generalization capabilities.

S2 - The main paper's diagrams (Fig. 1, 2) clearly illustrate both the core motivation—the need to bridge the classification/retrieval gap—and the architecture of the proposed methodology. The supplementary visualizations provide insight beyond raw metrics. The spatial error maps (Fig. 8), for example, visually demonstrate where and why the hybrid model outperforms its baselines, offering a powerful argument for its design. Furthermore, figures like the PCA visualization of prototypes (Fig. 14) give evidence that the model is learning meaningful, semantically grounded geographic features (e.g., rivers, forests), not just performing a black-box mapping. The paper openly discusses performance drops in rural areas and the use of proprietary Google StreetView data.

S3 - The leap from previous state-of-the-art, which operated at city or region levels, to a 10-country, continent-level scale is significant. The ability to achieve >50% recall at 100m precision at this scale is important. This has implications for real-world applications like autonomous navigation, global-scale augmented reality, and planet-scale digital twin creation. The demonstrated 30-60x reduction in database size compared to traditional methods makes such systems practically feasible.

S4 - Classification and cross-view retrieval are existing concepts. The authors' approach of learning ground-view “prototypes” and then fusing them with aerial embeddings to create a unified, compact “cell code” appears to be original. This synergy overcomes the individual limitations of each approach (imprecision in classification, scalability issues in retrieval).

W1 - Limited reproducibility: The most significant weakness is that the research relies on a massive, proprietary dataset (Google StreetView) and secondly, the authors state they will not release the pre-trained model weights, while they provide a reasonable ethical justification for not releasing the model. Still, this is a major shortcoming of the work.

W2 - The training requires massive computational resources (128 TPUv2s) making it inaccessible for most researchers to replicate or build upon from scratch.

---

> ### Author Rebuttal · Authors · 2025-07-31
>
> We thank the reviewer for their detailed and constructive review. We appreciate that they acknowledge a "leap from previous state-of-the-art" in terms of scale, and that our hybrid method "appears to be original" and that its "synergy overcomes the individual limitations of each approach". The reviewer kindly points to a "strong supplementary material" and experiments that suggest "strong generalization capabilities" to "completely unseen geographic areas" and "a different data domain (pedestrian-view Trekker dataset)". We address their questions and suggestions to improve the paper.
>
> _**Limited reproducibility: The most significant weakness is that the research relies on a massive, proprietary dataset (Google StreetView) and secondly, the authors state they will not release the pre-trained model weights, while they provide a reasonable ethical justification for not releasing the model. Still, this is a major shortcoming of the work.**_
> We acknowledge these concerns. We are however unable to release the data, since we do not own it, nor the models weights, due to privacy concerns. We however commit to releasing the training and evaluation code to make reproduction of this work easier. The supplementary material provides detailed statistics on our data distribution, data processing steps, and implementation details, which make it possible to reproduce our method on a different (sufficiently large) dataset. Our focus is on demonstrating how far we can push the envelope in large-scale geolocalization given sufficient data and resources.
>
> _**The training requires massive computational resources (128 TPUv2s) making it inaccessible for most researchers to replicate or build upon from scratch.**_
> We note that our models are trained on TPUv2, an architecture that was originally released in 2017 (the latest architecture is TPUv6e). With the scalable implementation described in the paper, our model can be trained on the smaller BEDENL dataset (used in the majority of experiments) with only 16 TPUv2, which are equivalent to 4 A100s: this is comparable to the VRAM requirements of recent works [1]. Exactly reproducing our results requires a setup equivalent to 26 A100 GPUs with 80GB VRAM: we acknowledge that this is large, but not necessarily out of reach for academic research labs.
>
> _**Could the authors provide a learning curve showing how the final model performance (e.g., Recall@1@100m) varies with the size of the training dataset?**_
> We agree with the reviewer that this ablation would provide valuable insights, and ease the barrier on reproducing our results. There are two ways to reduce the size of the training dataset: we can either (a) reduce the _total number of cells_ (i.e., geographical coverage), or (b) reduce the _density of images within a cell_, while maintaining the spatial coverage.
> An ablation like (a) is analogous to the BEDENL split we use for a large part of our evaluations. Recall is comparable to EuropeWest, suggesting that performance remains robust when increasing the size of the database. This can be attributed to the global discriminativeness of the prototypes. However, the impact of global negatives from classification will diminish as the map size is reduced, as it becomes more likely that hard negatives are part of the batch.
> As for (b), it is implicitly covered by Figures 3 (main paper: "recall vs training data density") and 7 (supplementary: "localization vs data density"). Classification is susceptible to the density and diversity of ground samples because it learns to remember content observed in each cell, and we would therefore expect a significant performance drop when significantly reducing data density. Cross-view retrieval is less sensitive to this because it is forced to encode visual information from the aerial tiles, and our hybrid method is able to combine the best of both modalities.
>
> Consequently, we claim that our approach can provide substantial improvements even at a smaller scale, thanks to its hybrid nature. This increased robustness is what enables us to run on more cells, with a lower density of images per cell.
>
> _**Are failures modes due to visual ambiguity, lack of data, or aerial-ground mismatch?**_
> This is difficult to answer, as we can only partially back up our intuitions with quantitative data. Figs. 4 and 8 clearly show that our method mostly fails in rural areas with low temporal coverage. In such cases, the method largely relies on cross-view retrieval. Visual ambiguity is also a problem, especially given large occluders, such as the truck covering almost the entire image in Fig. 13 (bottom row): these cases are very difficult to solve. The increased performance on highways (which have better data coverage) suggests that sparsity is the larger problem. Aerial-ground mismatches from the large domain gap were significantly reduced by our method thanks to the hybrid training and global discriminativeness.
>
> _**Could confidence estimation or uncertainty modeling help reject low-quality predictions?**_
> Such predictions could certainly help in filtering false-positive predictions, e.g. unlocalizable images. Implicitly, the ViT already learns which parts of the image are useful for localization. Adding confidence estimation and / or uncertainty modeling during training could help to further downweight unlocalizable images, which yield noisy gradients. During evaluation, the similarity score could be used as a prediction confidence. We believe that all these topics are exciting future research directions.
>
> _**Could the authors clarify whether the "Prototypes-Only" model was trained independently, or if its results were obtained by omitting aerial features at inference time from the full hybrid model?**_
> Please let us clarify: _Ours (cell prototypes)_ and _Ours (full)_ are two different models, trained independently. We evaluate them under different conditions: in Table 1-a ("ground retrieval"), we use either model to do VPR. In Table 1-c ("cell prototypes"), _Ours (full)_ omits the aerial features at inference time, and in Table 1-d ("hybrid") we use the aerial features. We will make this clear in the revision.
>
> _**[Continued:] ["Ours (full)" being better than "Ours (cell prototypes)" in Table 1-c] would suggest that joint training with aerial supervision improves prototype quality, even when not used at test time, which would be an interesting insight worth highlighting more directly.**_
> Correct. This is backed by experiments: the ablation study in Table 7 (supplementary) clearly shows that adding ground-aerial contrastive supervision during training greatly improves the recall rate with just prototypes (i.e., discarding aerials at test time). The aerial data smooths and stabilizes the prototypes, making them more robust to sparsity, both spatially and temporally. We thank the reviewer for pointing this out and we will highlight this insight more prominently in the revision.
>
> _**Could the authors comment on the feasibility of scaling the method to an entire continent (e.g., all of Europe) or even globally?**_
> We performed most of our experiments on 128 TPUv2 (i.e., older-gen) to keep experimentation reasonably fast, and run an extensive set of ablations. With the training recipe and scalable implementation described in the paper, it is possible to run all experiments with half the amount of TPUs on EuropeWest, and just 16 TPUv2 for BEDENL, which was our primary dataset during exploration.
> Consequently, without any adjustments, the method could scale up to twice the area of EuropeWest (basically all of Europe, excluding Russia). Note that this of course depends on how the data is distributed: some countries are more densely sampled than others.
> Scaling up beyond that is tricky as the amount of descriptors stored per device scales linearly. Potential techniques to mitigate this include parameter sharing (e.g. setting the first $k$ dimensions of all descriptors in a country identical), multi-stage training (e.g. alternating between cross-country classification and in-country localization), or low-dimensional projections of cell descriptors. For the latter, one could encode prototypes with a location encoder (similar to GeoCLIP [11]), yet provide the network some flexibility to implicitly store / remember information per cell explicitly.

---

### Official Review · Reviewer_bt81 · 2025-07-03

**Clarity:** 3
**Significance:** 2
**Originality:** 3
**Rating:** 4
**Confidence:** 4

**Summary:**

The paper presents a new approach to large-scale visual localization. It
combines recent advances in map cell prototype learning with cross-view matching
using overhead imagery.

The present paper seems to be the first one to combine the learning of cell
prototypes with overhead imagery. Specifically, the approach uses a form of
metric learning to optimize three kinds of embeddings: ground-level image
embeddings, overhead image embeddings, and map cell prototypes. At inference
time, all queries are assumed to come as ground-level images. By training cell
embeddings to function as database anchors, the method scales to extremely large
datasets without the need to build a massive database of ground-level images.

The analysis in the paper focuses on a novel in-house dataset and demonstrates
the scalability of this approach on very large scale coarse outdoor global
localization (coarse = most experiments consider a result to be a true positive
if it's within 200 m of its true location). The proposed method outperforms
several baselines while using less memory---the main bottleneck when performing
retrieval-based localization over large areas.

**Questions:**

## Questions
- [Q1] Is the Trekker dataset publicly available? The provided reference on the
  Google blog describes the sensor kit, but doesn't specify any dataset.
- [Q2] Thank you for adding details on avg. edge length when talking about S2 cells.
  It's helpful to contextualize the scale we are learning at in each tile.
- [Q3] In Eq. (1) it would be helpful to define the Q/A/P superscripts.
## References
- `[@faiss-bench]: https://github.com/facebookresearch/faiss/blob/main/benchs/README.md`
- `[@v2025bang]: https://arxiv.org/pdf/2401.11324v3`

**Ethical Concerns:**

["NO or VERY MINOR ethics concerns only"]

**Final Justification:**

I would like to thank the authors for their detailed rebuttal. I think their
clarifications regarding motivating use cases beyond robotics, code release, and
a promise of more lower-dimensional embeddings at least partly address all the
key questions and weaknesses I identified during my original review. The paper
is thoroughly written and I think it will make a strong contribution to the
field visual place recognition.

In conclusion, based on the author rebuttal and the arguments made by the
other reviewers, I would like to raise my rating to a weak accept.

**Limitations:**

yes

**Quality:**

3

**Strengths And Weaknesses:**

## Strengths
- [S1] The paper presents a novel architecture and training scheme for visual
  localization at continent scale. By supporting a customizable cell size at
  training time, the method allows for flexible trade-offs between higher efficiency
  (larger cells, less memory) and higher accuracy (smaller cells, more memory).
- [S2] Detailed quantitative analysis, including careful ablation study of key
  design decisions like the map cell granularity and the feature vector
  dimensionality. The supplementary material is very detailed, encompassing yet
  more ablations, lots of implementation details, as well as qualitative
  results.
- [S3] The proposed dataset is extensive and its train/eval split is temporal,
  which is representative of many real-world use cases.

## Weaknesses
- [W1] I think it is necessary to motivate the problem setting a bit more in the
  paper. While there is value in pushing cross-view matching accuracy further, I
  think applying it exclusively to outdoor visual place recognition is an overly
  artificial use case. On the one hand, the method assumes there is no GPS
  available to bootstrap the estimate. On the other hand, the method relies on
  high-quality aerial imagery for the cross-view matching component. Wouldn't
  the areas which are clearly visible in aerial imagery be precisely those ones
  with good GNSS availability? GNSS receivers have been getting very cheap and
  can routinely connect to multiple constellations (Galileu, BeiDou, GLONASS),
  making GNSS data an inexpensive commodity.
- [W2] The dataset cannot be released due to a licensing issue. This will make it
  difficult to reproduce the paper given the relatively unique problem setting,
  which requires ground-level panoramas in addition to aerial imagery.
- [W3] "Standard image retrieval techniques are simply not applicable [due to >100M
  images]" I think this statement requires some clarifications. Dimensionality
  reduction can be helpful in reducing memory footprint, while approximate
  nearest neighbor search can be very fast on modern systems while still
  achieving recall rates of 0.9 @ 10 or similar for 1B+ datasets.
  For example FAISS can run queries on the CPU in under 100 ms [@faiss-bench],
  while GPU throughput exceeds 500 queries per second [@v2025bang; Table III].
  - It would be informative to extend the "ground retrieval" section of Table 1
    a little. One direction could analyze certain variants of SALAD yielding
    lower-dimensional outputs. Another direction would be the aforementioned
    ANN, or something simpler, such as projecting high-dimensional embedding
    vectors to a lower-dimensional space using something methods like PCA.
- [W4] (Minor) A more detailed explanation of how localization works as "a
  combination of aerial and cell code retrieval" (cf. L210) would be helpful, as
  right now it isn't immediately clear to me what this means mathematically.

---

> ### Author Rebuttal · Authors · 2025-07-31
>
> We kindly thank the reviewer for the insightful and constructive feedback. The reviewer acknowledges the "novel architecture and training scheme for visual localization at continent scale", supported by a "detailed quantitative analysis, including a careful ablation study of key design decisions". We also appreciate the comment stating that "the supplementary material is very detailed", recognizing that we invested extra time and effort. We thank them for suggesting improvements in the readability and motivation of the approach (with which we agree).
>
> _**I think it is necessary to motivate the problem setting a bit more in the paper.**_
> We agree that the motivation section should be reworded, and that use-cases such as robotics or augmented reality are not a great fit. However, global, prior-less geo-localization is an important problem, enabling us to localize any images without GPS, such as older / historical images, videos (which usually lack a GPS tag), or images where the EXIF metadata was accidentally stripped, i.e. after image transformations. It can be applied to arbitrary imagery, such as images distributed in the media for validation, and enables criminal investigations or could support the detection of AI generated images. Finally, models trained for large-scale geolocalization can learn high-level semantics (geographical/cultural patterns, see Fig. 14) and could be a valuable pre-training for other tasks (though this is outside the scope of our work).
>
> _**The dataset cannot be released due to a licensing issue. This will make it difficult to reproduce the paper given the relatively unique problem setting, which requires ground-level panoramas in addition to aerial imagery.**_
> We acknowledge these concerns. We are however unable to release the data, since we do not own it, nor the models weights, due to privacy concerns. We however commit to releasing the training and evaluation code. Alongside the detailed description of the data distribution and processing steps in the supplementary, we believe that this should make it possible to reproduce our work on a different (sufficiently large) dataset.
>
> We note that ground-level panoramas are not strictly required. We stitch the images into panoramas and sample crops from them as a simple data augmentation technique, to generate a wide variety of view points and focal lengths. We believe that sufficient diversity in standard imagery would work equally well. Our requirements on aerial imagery are less strict than Fervers et al [29], where the best model was trained on 20cm resolution tiles, compared to 60cm for our method.
>
> _**[...], the method relies on high-quality aerial imagery for the cross-view matching component. Wouldn't the areas which are clearly visible in aerial imagery be precisely those ones with good GNSS availability?**_
> While it is true that these use cases overlap, there are plenty of tasks / data sources where GPS is not available at all, see discussion above.
>
> _**Dimensionality reduction can be helpful in reducing memory footprint, while approximate nearest neighbor search can be very fast on modern systems while still achieving recall rates of 0.9 @ 10 or similar for 1B+ datasets.**_
> We agree with the author that the mentioned statement did not take into account recent advancements in efficient image retrieval applicable during evaluation, and we will tone this down. However, the compression in database size via prototypes would also increase throughput and reduce memory when paired with dimensionality reduction and/or ANN. Furthermore, such techniques are not applicable during training, while we show that compression through cell prototypes reduces memory to a point where we can utilize global negative proxies for supervision, which boosts accuracy.
>
> We contacted the authors of SALAD and asked for variants with lower dimensionality, however, these models are not available anymore. For the final paper version, we consider adding another baseline with smaller descriptors such as EigenPlaces [18] or CosPlace [17], or SALAD with reduced dimensionality if the authors can retrain the models. However, as illustrated in Table 9 (supplementary), we expect them to be significantly weaker than our classification-only baseline evaluated for VPR.
>
> _**A more detailed explanation of how localization works as "a combination of aerial and cell code retrieval" (cf. L210) would be helpful, [...].**_
> We agree that this section could be clearer. we train the network to find the optimal cell $j*$ by maximizing the similarity between the query and the representations in each modality: $j^* = argmax_{j} (z_i^Q)^T z_j^A$ and $j^* = argmax_{j} (z_i^Q)^T z_j^P$. To combine the search in both modalities, we look for the cell $ j^* $ with the highest combined similarity, which is achieved by summing the individual similarities: $j^* = argmax_{j} ((z_i^Q)^T z_j^A + (z_i^Q)^T z_j^P)$. Due to the linearity of the dot product, this is equivalent to first summing the modality-specific representations of the cell and then calculating the similarity with the query: $j^* = argmax_{j} (z_i^Q)^T (z_j^A + z_j^P)$. If we define $z_j^{CELL} = z_j^A + z_j^P$, then the problem becomes $j^* = argmax_{j} (z_i^Q)^T z_j^{CELL}$. Note that all embeddings $z$ are L2-normalized here. We will use some of the extra page afforded in the revision to clarify this, adding an additional figure outlining our training and inference pipeline.
>
> _**Is the Trekker dataset publicly available?**_
> The Trekker dataset is not publicly available, as it has the same legal requirements as the StreetView data used for training. As such, it is subject to takedown requirements, which need to be honored in a timely manner after being triggered. This requires a significant infrastructure investment, as the data owner needs to ensure that all copies are deleted.
>
> We acknowledge that there are alternative ways to make the data "available": for instance, platforms like Kaggle or HuggingFace allow submissions on hidden test sets, and we are currently evaluating this type of scenario. This is, however, not without problems: these competitions still typically require a fair amount of public "training" data, or participants must resort to blind leaderboard probing.
>
> _**In Eq. (1) it would be helpful to define the Q/A/P superscripts.**_
> Indeed this would improve readability, and we acknowledge that this was a point of confusion with multiple reviewers. We will incorporate this feedback in the revision and improve the explanation of Eq. (1).

---

> > ### Comment · Reviewer_bt81 · 2025-08-06
> >
> > I would like to thank the authors for the detailed rebuttal. I appreciate the
> > details and I apologize for the relatively late follow-up.
> > - *Motivation.* Thank you for clarifying the additional use cases. I agree that
> >   localizing historical data can be very valuable, and using localization to
> >   detect AI-generated images is a very clever application I hadn't thought of
> >   before! I suppose my comment was coming from a(n overly narrow) robotics
> >   perspective. These are valid use cases and I think they definitely lower the
> >   weight of my original [W1]. Mentioning these use cases in the introduction can
> >   strengthen the paper and motivate it to a broader audience, so if there is
> >   space, I would encourage the authors to discuss them in a sentence or two.
> > - *Dataset.* A code release would definitely help improve reproducibility, but I understand the challenges, having been in a similar position myself in the past.
> > - *Dimensionality Reduction / ANN.* Thank you for the explanations and for
> >   considering the additional baselines. I think these additional experiments
> >   would be nice from an engineering / scalability perspective, but I don't think
> >   they are critical to the paper.
> > - *How Localization Works.* Thank you for the additional explanation, it makes
> >   things much clearer in my head! Even if it doesn't fit in the main paper,
> >   having this step-by-step explanation in the supplementary would be very
> >   helpful to future readers.
> > - Thank you also for the other clarifications, like regarding the Trekker
> >   dataset and the promised notation improvement.
> >
> > I would like to thank the authors for their detailed rebuttal. I think their
> > clarifications regarding motivating use cases beyond robotics, code release, and
> > a promise of more lower-dimensional embeddings at least partly address all the
> > key questions and weaknesses I identified during my original review. The paper
> > is thoroughly written and I think it will make a strong contribution to the
> > field visual place recognition.
> >
> > In conclusion, based on the author rebuttal and the arguments made by the
> > other reviewers, I would like to raise my rating to a weak accept.

---

### Official Review · Reviewer_7jVE · 2025-07-03

**Clarity:** 3
**Significance:** 4
**Originality:** 3
**Rating:** 5
**Confidence:** 4

**Summary:**

Authors present a novel method for geo-localization that combines cell prototypes and aerial embeddings to operate at continent level. Standard Visual Place Recognition (VPR) requires a very large dataset to work at continent level, whereas cross-view (ground - aerial) retrieval and cell classification lack accuracy. They propose to learn all modalities at the same time in a hybrid approach such that ground-aerial, aerial-cell, and ground-cell relations are all considered. Then they create the final cell prototypes by adding the cell and the aerial embeddings. Exhaustive evaluation shows that the method is accurate and scalable.

**Questions:**

- Can the authors give details or intuitions on why the retrieval works on the sum of prototypes and aerial descriptors, even though the norm of the descriptor changes in that case?
- The model is never trained to match ground-ground descriptor and the training loss does not include a component that brings together two similar ground locations. However the model achieves outstanding metrics on the VPR modality. Can the authors give details or intuitions about this?
- Can the authors give more details on what Figure 5 (main paper) and Figure 12 (supplementary) represent? In figure 5 I assume we are seeing the query image, and the rank represents in which rank position the correct cell + aerial descriptor was ranked? Figure 12 is hard to understand, is the a) input an aerial image? It finds the query images for different methods that result on the same retrieved area (something like reversed retrieval)?
- Table I (c), what’s the difference between Ours (cell prototypes) and Ours (full)? Is the training process where Ours (cell prototypes) ignores the aerial edges, and ours (full) is the full training but not summing the aerial descriptors?

**Ethical Concerns:**

["NO or VERY MINOR ethics concerns only"]

**Final Justification:**

After reading the authors rebuttal, discussing further with them some of the points, and reading the points made by other reviewers I keep my original score and I am willing to accept the paper.

In general is a technically sound, well written and novel paper. It's main weakness is the lack of reproducibility and the limited contribution to the community if data and weight arent released. However I believe strengths overweight these issues.

**Limitations:**

The authors detail some of the limitations scattered around the paper. However a reader would appreciate a limitations section where all limitations are compiled together and discussed.

Regarding the societal impact, authors discuss it in the supplementary. This family of geo-localization methods have privacy concerns. The authors won’t publish the model weights nor the dataset, so these concerns are minimized. However,  as a consequence the utility and relevance for future work will be affected by the lack of public weights.

**Quality:**

4

**Strengths And Weaknesses:**

Strengths:

Addressing the scalability of visual geo-localization systems is a very interesting topic for the community.

The paper suggests that previous commonly used approached like ground-ground retrieval (VPR) and cross retrieval do not scale well or are not accurate. Authors also provide exhaustive experimentation for these claims.

Authors present a conceptually simple yet powerful hybrid approach. Combining the cell prototypes with aerial descriptors is, as far as I know, a novel idea, and it’s a sensible one. Cell prototypes and aerial descriptors have different qualities (ground has very local details but miss on density, aerial has the density of the whole area but it’s more difficult to match). These different qualities combine very well with each other, so merging them is a reasonable idea.

The training pipeline is interesting as it proposes a novel way of training ground, aerial and cell prototypes at the same time by considering the edges that connect all of these modalities. Regarding training, it is also appreciated the significant engineering effort to train the model on such a scale, from collecting all the training assets, to distributing the training across machines.

The evaluation is very complete. Authors present results (mainly) on two customs datasets and EuropeWest, and provide extensive results evaluating different modalities with different state-of-the-art techniques. Their results are very solid, obtaining very good metrics with a reasonable dataset size. Their introspective and ablative experiments. both in the main paper and supplementary, are very complete and interesting. They help the reader address the importance of the different components as well as its strengths and limitations.

Weaknesses:

Although the paper is, in general, very well written and easy to follow, this reviewer would appreciate more explicit or detailed information about the training and inference process. As it is just detailed on text, it is not fully clear until the text is read a few times how the training and inference works. This reviewer would appreciate an algorithmic representation (perhaps supplementary) of how this works.

The paper proposes to sum cell prototypes and aerial descriptors to obtain the final cell descriptors. However, the model is not trained to match the sum of these descriptors. Although the retrieval works, readers would appreciate a theoretical justification on how the retrieval on the sum space keeps working, or provide cites of papers handling this in a similar way.

They train on images from 2017,18,19,20,21,22,24 and evaluate on 2023, whereas a more realistic situation would be to train on 2017-2023 and evaluate on 2024. It is expected to have past images for training, and then evaluate on the newer images without training on future images.

It is appreciated that the authors run their model as a ground-ground VPR model on Table I (a) to show the limitations of this approach. SALAD with its biggest descriptors is unreasonable to run (5TB), but what about SALAD with smaller descriptors? They also report results with 2049 descriptor size.

---

> ### Author Rebuttal · Authors · 2025-07-31
>
> We kindly thank the reviewer for the positive feedback, stating that the method is a "conceptually simple yet powerful hybrid approach", and highlighting that the proposed combination of aerial descriptors and cell prototypes is "novel" and "sensible". The reviewer also finds that the "training pipeline is interesting" and recognises the "significant engineering effort" and a "very complete evaluation". We address their questions.
>
> _**Although the paper is, in general, very well written and easy to follow, this reviewer would appreciate more explicit or detailed information about the training and inference process.**_
> We thank the reviewer for this valuable feedback. We agree that, while the method is conceptually simple, the paper may not make this sufficiently clear. We will use the extra page afforded by the final version to outline both training and inference in a new pipeline figure that should help convey this message. Notably, we will highlight (a) how aerial and cell prototypes are merged before inference, and (b) the different losses and modalities used during training, summarizing Eqs. (1) and (2).
>
> We also acknowledge that some terms may impair readability, e.g. "cell codes" and "prototypes" (which are the same thing).
>
> _**The paper proposes to sum cell prototypes and aerial descriptors to obtain the final cell descriptors. However, the model is not trained to match the sum of these descriptors.**_
> Indeed, we never train the network to match the (weighted) sum of aerial embeddings and prototypes. However, we train the network to find the containing cell in each modality, i.e. we learn: $j^* = argmax_{j} (z_i^Q)^T z_j^A$ and $j^* = argmax_{j} (z_i^Q)^T z_j^P$. To combine the search in both modalities, we look for the cell $ j^* $ with the highest combined similarity, which is achieved by summing the individual similarities: $j^* = argmax_{j} ((z_i^Q)^T z_j^A + (z_i^Q)^T z_j^P)$. Due to the linearity of the dot product, this is equivalent to first summing the modality-specific representations of the cell and then calculating the similarity with the query: $j^* = argmax_{j} (z_i^Q)^T (z_j^A + z_j^P)$. If we define $z_j^{CELL} = z_j^A + z_j^P$, then the problem becomes $j^* = argmax_{j} (z_i^Q)^T z_j^{CELL}$. Note that all embeddings $z$ are L2-normalized here. We will use some of the extra page afforded in the revision to clarify this, adding an additional figure outlining our training and inference pipeline.
>
> _**They train on images from 2017,18,19,20,21,22,24 and evaluate on 2023, whereas a more realistic situation would be to train on 2017-2023 and evaluate on 2024.**_
> The explanation is very simple: we started this project in late 2024, so we did not have a full year of data for 2024 at the time (StreetView data needs to be filtered and posed, which takes time). We chose 2023 for evaluation because it gave us the best possible coverage at the time, allowing us to evaluate on most areas we train on. We did not think to reevaluate this decision later on. We agree that we may somewhat overestimate the model's robustness against long-term temporal changes, but we believe this is a minor factor.
>
> _**SALAD with its biggest descriptors is unreasonable to run (5TB), but what about SALAD with smaller descriptors? (Table 1)**_
> We contacted the authors of SALAD, who declared that the weights of the more compact models (described in the SALAD paper) had unfortunately been lost.The model would thus have to be retrained. However, we refer to our evaluation in Table 9 (supplementary material) which shows that our retrieval baselines significantly outperform SALAD with large descriptors (D=8448). A smaller SALAD (D=2176) would likely be even weaker. (Please refer to our response to R1 / f1mU for more details.)
>
> _**The model is never trained to match ground-ground descriptor and the training loss does not include a component that brings together two similar ground locations. However the model achieves outstanding metrics on the VPR modality. Can the authors give details or intuitions about this?**_
> Classification serves as a proxy task for image retrieval, as it puts all image embeddings belonging to the same class close to each other [17]. As we note in the paper (L81), some of the strongest image retrieval models are trained with a classification loss [17, 18].
>
> _**Can the authors give more details on what Figure 5 (main paper) and Figure 12 (supplementary) represent?**_
> We apologize for the lack of clarity, which we will improve in the final version of the paper.
>
> Fig. 5 shows 36 query images sorted by their rank, which describes the position of the first positive cell (within 200m of the query) in the list of database cells sorted by descriptor similarity (from the hybrid database). In other words, a query with rank K has K-1 false positive predictions.. A larger rank corresponds to a lower localization accuracy. Most queries with low rank (but not all) contain a fair amount of structure, whereas images showing mostly trees or sky are difficult to localize.
>
> Fig. 12 illustrates the localization error obtained from different variants of our model. We show a ground-level query (which is repeated 3 times), and the aerial tile for the cell the query belongs to in a). The 3 columns (prototypes, aerial, full) denote the model used for retrieval, with “full” being the hybrid (aerial + prototypes) database. Overlaid on each query image, we report the distance to the top-1 nearest neighbor (i.e., its error). The border colors help illustrate this error, where green means low error. Calling the first column "input" is misleading, we will correct this (e.g. replacing it with “cell”).
>
> _**Table I (c): what’s the difference between Ours (cell prototypes) and Ours (full)? Is the training process where Ours (cell prototypes) ignores the aerial edges, and ours (full) is the full training but not summing the aerial descriptors?**_
> Yes, _Ours (cell prototypes)_ is a pure classification method without aerial edges, and _Ours (full)_ is the full training (with aerial edges), but ignoring aerial descriptors during evaluation. We will try to highlight this setup in the final paper version.

---

> > ### Comment · Reviewer_7jVE · 2025-08-02
> >
> > Thanks to the authors for their detailed answer.
> >
> > I appreciate all clarifications, specially on why the model works on the sum of descriptors. I suggest including some of these clarifications on the final version to help readability.
> >
> > Regarding the 2024 training data: I understand how you ended up in such situation, but I think the evaluation would be fairer without the 2024 training data.
> >
> > Regarding the SALAD results and suggestion to use smaller descriptors. To overcome the lack of smaller descriptor's weights:
> >  1) Run the experiment by chunks. Although I understand the point of showing OOM to exemplify the magnitude of the database, you could compute the evaluation without storing all descriptors at once. I) Save descriptors for a chunk of database images. II) Compute distances to query descriptors III) Save distances. Repeat for all chunks, and you only need to save all the distances but not all the descriptors. This is slow to compute, and computationally more expensive, but from all experimentation in the paper it looks like resources are not a big limitation for the authors ;)
> >
> >  2) As other reviewers suggested you could also use dimensionality reduction.

---

> > > ### Author Response · Authors · 2025-08-05
> > > **Official Comment by the Authors**
> > >
> > > We thank reviewer 7jVE for the suggestions to improve our work.
> > >
> > > **Regarding the 2024 training data:**
> > > We acknowledge that the current split cannot fairly and exhaustively evaluate the robustness to future temporal changes. We however note that, in many locations, the database images were captured long before 2023, so this scenario is already partially covered by our evaluation. Instead of retraining all models on data including  2023 and evaluating on 2024, we will evaluate new queries captured in the first half of 2025, and compare them to 2023 queries in locations where data from both 2023 and 2025 is available. This will better demonstrate the model's robustness to temporal changes.
> > >
> > >  **Regarding the SALAD results and suggestion to use smaller descriptors:**
> > > We thank the reviewer for the valuable suggestions, and we agree that this is technically possible. It does require some engineering effort, and we will make a best-faith effort to provide numbers for SALAD in the main table.

---

> > > > ### Comment · Reviewer_7jVE · 2025-08-07
> > > >
> > > > Thanks for your intention on doing these experiments.
> > > >
> > > > The paper is technically good and these experiments may improve it further more. Anyway, I am happy to accept the paper in its current form. If these experiments are finished due time, please incorporate them to the final text.

---

### Official Review · Reviewer_f1mU · 2025-07-05

**Clarity:** 3
**Significance:** 3
**Originality:** 4
**Rating:** 4
**Confidence:** 4

**Summary:**

This paper addresses the problem of large-scale fine-grained geolocalization. The main idea is to build a method that provides a good trade-off between precision and scalability by combining proxy classification learning with cross-view retrieval. It leverages contrastive learning to jointly train ground-level, aerial, and cell prototype embeddings via a modified multi-similarity loss, allowing effective localization even with sparse ground data. The aerial imagery embeddings are fused with the learned cell prototypes to generate cell embeddings and build a scalable reference database for location estimation. At inference, this enables retrieval of the top-ranked database entries/cells given ground-level query images, achieving fine-grained localization with very high accuracy across extensive geographic regions covering almost an entire continent. Also, a large-scale dataset of about 150M images sourced from Google StreetView and open-access aerial resources is composed for the training and evaluation of the proposed method, covering a large part of Western Europe. The proposed approach outperforms existing methods at the continental scale, achieving an excellent trade-off between accuracy and complexity. The generalization of the method is also benchmarked on unseen areas during training, and on different domain data, i.e., ground view images captured from backpacks.

**Questions:**

Please refer to the review where several questions are provided.

**Ethical Concerns:**

["NO or VERY MINOR ethics concerns only"]

**Final Justification:**

Most of the concerns raised have been addressed in the authors' rebuttal. However, the most important one remains unaddressed: the comparison with SALAD trained with the same dataset is missing, which represents the SotA in VPR. This is the only way to assert that the current retrieval-based methods do not significantly outperform the proposed hybrid method when trained on the same scale of data. Additionally, classification-based baselines implemented with a finer-granularity grid that matches the one effectively used by the proposed method should be evaluated for fair comparison. To this end, the original rating is maintained.

**Limitations:**

yes

**Paper Formatting Concerns:**

No issue with the formatting is spotted.

**Quality:**

3

**Strengths And Weaknesses:**

**Strengths**
* The novelty of the work is high. The paper introduces a novel hybrid approach that combines ground-level and aerial imagery for image geolocation in a unified retrieval framework. Unlike prior work, this fusion enables leveraging complementary information from both views to achieve fine-grained localization. Additionally, the proposed training uses multi-similarity loss, employed also in [23]; yet, it is modified to serve the setup of the proposed approach as it aligns the embeddings of ground images, aerial tiles, and learned spatial prototypes through a joint loss.
* A very large-scale dataset is assembled from high-quality sources, spanning most of Europe and covering ten countries and hundreds of thousands of square kilometers. This dataset far exceeds the geographic scope of existing benchmarks and enables meaningful evaluation of geolocation methods at a scale relevant for real-world deployment.
* The proposed method demonstrates very high performance, achieving robust localization accuracy even at top-1 recall within several granularity ranges. This level of precision at such a coarse geographic scale is a significant advancement over prior methods, which typically operated only at city-level or regional granularity.
*  Despite high accuracy, the proposed approach offers a good trade-off between performance and computational cost. The proposed hybrid method achieves very competitive performance comparable with the method's performance when operating as ground retrieval, while requiring an order-of-magnitude smaller database. It achieves up to 30$\times$ reduction in storage with a sacrifice of about 3% in performance, making it suitable for large-scale deployment.
* The generalization capability of the proposed method is evaluated by applying the trained model to new geographic regions (e.g., the UK and Ireland) without retraining by using only aerial imagery for the new region. The model can still localize a good portion of queries, demonstrating that its learned representations are not fully tied to the training region and can transfer across different geographies. Also, cross-domain generalization is tested based on street view queries from backpacks, where the model shows good robustness.

**Weaknesses**
1. The SALAD baseline is not fairly represented. It is reported as OOM for the large-scale evaluation, but this is left unresolved. A fairer comparison would be either training SALAD with lower-dimensional descriptors, as done for the proposed approach, or applying approximate nearest neighbor search (e.g., via FAISS) via quantizing and/or indexing to avoid the brute-force bottleneck. What is the status (and potentially the upper bound) of the current state-of-the-art?

2. It is unclear whether baselines are treated in the same way as the proposed approach. Table 1 shows that classification-based baselines underperform significantly, which is somewhat surprising. It is unclear whether they were trained and tuned in the same way as the proposed model. Also, could using a different resolution (e.g., cell level L=16) or other factors potentially improve or reverse the poor performance?

3. Comparisons with more baselines that follow different training or inference approaches would provide a more complete picture of the current state-of-the-art. For example, methods with alternative training paradigms like GeoCLIP [11] or the diffusion-based method in [15] can be included. These methods operate under different principles but aim for the same geolocation task. Additionally, hybrid methods [A, B] that combine classification and retrieval, targeting high performance on low-granularity ranges.

4. The impact of the work would be increased if more cross-domain/cross-dataset evaluation is provided based on popular datasets.
The paper primarily evaluates on regions and data sources similar to the training distribution (Google Street View in Western Europe). To better establish generalization, it would be beneficial to include results on widely-used geolocation datasets such as IM2GPS3k or YFCC4k, which come from a very different distribution (e.g., social media photos, diverse cameras, global coverage). Even if full coverage is not feasible, experiments using subsets of those datasets overlapping with the training region or using coarse aerial sampling over testset locations could strengthen the case for robustness across domains.

5. The selection threshold of negative prototypes $\mathcal{N}(i)$ is missing. In L144-145, it is mentioned that a negative prototype threshold, which defines which spatially distant cells contribute to the loss, is applied. This threshold controls the selection of hard negatives and may significantly impact learning. What is this threshold value, and how does it affect training? Also, the sensitivity analysis of hyperparameters $\alpha$, $\beta$, $\lambda$ could be provided, since hyperparameter tuning has already been performed.

6. In Eq. 2, the loss includes two similarity terms between the ground and aerial embeddings, i.e., $z^{Q \top}_i z^A_j$ and $z^{A \top}_i z^Q_j$​). Since the latter will be optimized when image $j$ is used as the anchor, is it necessary to include it? Does it impact the training, or could it be removed to simplify the loss?

7. Aerial image source and characteristics are underspecified. A mix of aerial and satellite imagery at high resolution is used, but no sufficient details are provided about the source and the combination of the data. Since aerial imagery varies widely in style and quality depending on the provider or region, a more detailed description would help assess whether the method generalizes beyond this specific imagery type.

8. ArcFace/CosFace are widely-used losses employed in the field [17, B]. It would increase the impact of the paper if the performance of the proposed approach trained with such losses is assessed.

9. Reproducibility remains a concern. Although the authors acknowledge the use of proprietary data (e.g., large-scale Google Street View imagery) and extensive compute resources (e.g., TPUs), this still limits reproducibility. Without access to the dataset or an open-source equivalent, and given the infrastructure needed to scale training to millions of prototypes, it will be difficult for the community to verify or build upon this work without significant effort. Providing the code would mitigate the effort needed for reproduction, but other major details in the process (mainly the dataset generation) might be intractable. It would be very beneficial if the authors could provide some way for the community to submit solutions (e.g., code and trained model) for the evaluation on the composed dataset.

[A] Leveraging EfficientNet and Contrastive Learning for Accurate Global-scale Location Estimation. ICMR, 2021\
[B] Divide&Classify: Fine-Grained Classification for City-Wide Visual Geo-localization. ICCV, 2023

---

> ### Author Rebuttal · Authors · 2025-07-31
>
> We appreciate the reviewer's warm reception of our paper, who recognized its "high" novelty, scope ("enables meaningful evaluation of geolocation methods at a scale relevant for real-world deployment"), ("very high") performance, applicability ("suitable for large-scale deployment"), and generalization capabilities (to a subset of modalities or new viewpoints). We answer their questions.
>
> _**What is the status (and upper bound) of the current state-of-the-art?**_
> Please note that we flag the top row in Table 1 as OOM primarily to drive home the point that retrieval is an inefficient approach to solve geolocalization at this scale.
> Moreover, in Table 9 (supplementary) we show that pre-trained SALAD is significantly weaker than our best model in the VPR evaluation, even out-of-domain (i.e. cross-area). In-domain, this gap would only increase. Thus, SALAD with FAISS / ANN would in all likelihood be a weaker baseline than our best VPR method.
> We reached out to the authors of SALAD, but unfortunately the weights of the smaller SALAD variants are currently unavailable. Retraining SALAD on our datasets would be insightful, but this requires defining positive anchors i.e. spatially close pairs and careful engineering & tuning, which is beyond the scope of the rebuttal.
>
> _**It is unclear whether baselines are treated in the same way as the proposed approach. Table 1 shows that classification-based baselines underperform significantly, which is somewhat surprising.**_
> The classification baselines labeled as Hierarchical and Haversine rely on the cross-entropy loss, which we indeed found to significantly underperform the multi-similarity loss, on which the third baseline relies: _Ours (prototypes-only)_.Note that we consider it a _baseline_ because our main contribution is on _hybrid_ geolocalization. We will clarify this in the final version.
>
> All classification baselines were trained on the exact same data, and we carefully tuned them. We observed that standard softmax-based classification results in significant overfitting, which we were not able to resolve with a lower or learned temperature. Our intuition is that visual aliasing in combination with low data diversity leads to this behaviour. The network merely remembers "viewpoints" rather than learning more robust patterns. To reduce this, we mask out cells that are spatially too close to the query (L144), which improves accuracy. We also observed that our loss is more robust to unlocalizable images, likely because of supervising absolute- instead of relative similarities.
>
> More implementation details of our baselines are listed in the supplementary, section C.1.
>
> _**...methods with alternative training paradigms like GeoCLIP [11] or the diffusion-based method in [15] can be included.**_
> We thank the reviewer for suggesting additional baselines. We did re-implement GeoCLIP [11] during our research, but failed to obtain meaningful results with it. GeoCLIP was designed for much coarser localization (>10km) and its positional encoding is thus not suitable to capture high-frequency details required for fine-grained geolocalization. To alleviate this, we: (a) increased the size of the location encoder, and (b) replaced the positional encoding with SIREN [C]. Even then the method lagged behind the already weak classification baselines. While we believe that it should be possible to solve this problem, it requires a non-trivial amount of research and we opted to not pursue it further. We did not include these results in the paper because they are not fully-realized, but we will make a best-faith effort to further develop this baseline for the camera-ready if the paper is accepted.
>
> The diffusion-based approach by Dufour et al [15] is also designed for much coarser global localization and, like GeoCLIP, would likely require significant adjustments to work well in our setting.
>
> [A]: This work suggests a Search-within-Cell (SwC) scheme, where coarse classification is  refined with retrieval and clustering to database images within this cell. This is orthogonal to our approach.
>
> [B]: This work is closely related to ours, and we thank the reviewer for pointing it out. Notably, they also use a contrastive loss (ArcFace) for classification, which outperforms cross-entropy losses. This agrees with our findings. We applied ArcFace and CosFace on classification in an earlier stage of our project but we observed that they were not as effective as the multi-similarity loss (baseline _Ours (prototypes)_). Due to time and resource constraints, we are not able to rerun these experiments in the short rebuttal period but we will add them in the final version.
>
> [B] also introduces  a mixture-of-experts approach to mitigate visual aliasing. In our work, we avoid visual aliasing by masking out negatives that are spatially close to the query, which is conceptually simpler. The suggested "mixed pipeline", which performs retrieval on the images from the top-k classes (=cells), would also apply to our method, and is thus orthogonal.
>
> _**The impact of the work would be increased if more cross-domain/cross-dataset evaluation is provided based on popular datasets.**_
> Public datasets are designed for a very different problem: Finding patterns that are common in a certain geographic area (e.g. country, city). While such cues are powerful, they do not support fine-grained localization, i.e. finding the right corner of a street. Therefore, while the end goal is similar, comparing our method to public works on these benchmarks is of limited value.
>
> While we agree that further evaluation on a wider variety of real-world sensors and viewpoint conditions would be valuable, we believe the proposed datasets, IM2GPS3K and YFCC4K, are not suitable:
>
> i) They are noisy, with a very large ratio of unlocalizable images (faces, food, indoor scenes). Results would be hardly comparable without extensive manual filtering. Failures due to unlocalizable images or different intrinsics would be indistinguishable. Nevertheless, we consider adding an ablation of FoV vs. recall on our dataset to show the models robustness to varying intrinsics. .
>
> ii) The granularity gap with literature is too large, even for in-domain images, and the results would likely just highlight this fact, which is of little value. We believe that retraining public baselines on our (cleaner & larger) data, as we do in Table 1, is more interesting.
>
> iii) As the reviewer notes, our evaluation is thorough: we report the performance against viewpoint changes (Table 4), data sparsity & diversity (Table 3), geographical variations (Figs. 5 & 8), and even population density (Fig. 9), which clearly illustrate where our method improves or fails. We agree that alternative devices are missing in this evaluation, but to the best of our knowledge, there is no public dataset with sufficient quality and coverage.
>
>
> _**The selection threshold of negative prototypes is missing.**_
> We exclude any negative cells within $d=200m$ to address visual aliasing, i.e., queries that observe content over multiple cells. Without this threshold, the classification accuracy drops significantly. We apologize for this oversight, and will add a more extensive sensitivity analysis to the paper.
>
> _**In Eq. 2, the loss includes two similarity terms between the ground and aerial embeddings… Does it impact the training, or could it be removed to simplify the loss?**_
> Indeed, one of the terms is implicitly optimized from another anchor in the batch. Excluding this term would certainly change the relative weight between contrastive and global terms. We have not conducted an ablation of this on our full model, however, in the SALAD aerial baseline, we found that this "bidirectional" variant slightly boosts accuracy over the unidirectional variant (where the term $z_i^A$ is excluded). Fervers et al [29] also uses a bidirectional loss.
>
> **Aerial image source and characteristics are underspecified.**
> In general, we prefer the highest quality and most recent capture available for each location, determined by the geometric resolution of the aerial tiles. The imagery is owned by Google and not available externally. The orthophotos have a resolution of 10cm or 25cm (which we downsample to 60cm),  stitched from oblique views using a high-quality digital surface model. The data was captured over the last 10+ years, and is the same data visible on Google Maps / Google Earth at the highest zoom level. We will add these details to the paper.
>
> **Reproducibility remains a concern.**
> We acknowledge these concerns. We are however unable to release the data, since we do not own it, nor the models weights, due to privacy concerns. We however commit to releasing the training and evaluation code to make reproduction of this work easier. The supplementary material provides detailed statistics on our data distribution, data processing steps, and implementation details.
>
> **extensive [TPU] compute resources[...]**
> With the scalable implementation described in the paper, our model can be trained on the smaller BEDENL dataset with only 16 TPUv2 (an older-gen model originally released in 2017), which are equivalent to 4 A100. This is comparable to the VRAM requirements of recent works [1].
>
> **It would be very beneficial if the authors could provide some way for the community to submit solutions (e.g., code and trained model) for the evaluation on the composed dataset.**
> There are indeed ways to do this, as platforms like Kaggle or HuggingFace allow submissions on hidden test sets. We thank the reviewer for this suggestion, and we are actively evaluating similar scenarios. However, we note that these competitions still typically require a fair amount of public "training" data, or participants resort to blind leaderboard probing. This is difficult, as the data owners have to comply by law with takedown requirements.
>
> [C] Implicit Neural Representations with Periodic Activation Functions, Sitzmann et al, NeurIPS, 2020.

---

> > ### Comment · Reviewer_f1mU · 2025-08-04
> >
> > Thanks to the authors for the detailed and thorough rebuttal. The clarifications regarding scalability, methodology, and the limitations of other baselines (e.g., GeoCLIP, diffusion models) are appreciated.
> >
> > However, one key point remains insufficiently addressed: the quantitative comparison with SALAD under equivalent conditions. This is critical to substantiate the central claim that the proposed hybrid method outperforms retrieval-based approaches at scale. It remains unclear whether the observed performance gains stem from the method itself or from access to higher-quality, larger-scale data. In particular, we need to assess whether the superior performance stems from the proposed approach or the better quality/more data. While retraining SALAD during the rebuttal is understandably hard, this comparison remains essential and very interesting. Nevertheless, even a comparison with ANN (e.g., Product Quantization) or dimensionality reduction, generating descriptors of the same memory footprint, would be insightful.
> >
> > Additionally, several points rely on qualitative statements without quantitative evidence. For example, what is the performance when train with ArcFace/CosFace? What is the impact of the selection threshold and bidirectional loss? What are some indicative results of GeoCLIP and classification-based methods with other settings?
> >
> > Regarding generalization, the rationale for not evaluating on public datasets like IM2GPS3k or YFCC4k is acknowledged. However, reporting performance on a filtered or overlapping subset could still offer meaningful insight into domain robustness. Moreover, in the response to another reviewer, the authors mention that a key motivation for the method is *to be applied to arbitrary imagery*, which contradicts the earlier justification for excluding such datasets.
> >
> > Regarding reproducibility, the authors' commitment to release the training and evaluation code is appreciated. Yet, the lack of access to the data and pretrained models, combined with the significant computational requirements to train the full model, poses a high barrier. As suggested by the authors, offering a mechanism for external evaluation would be a valuable step and is strongly encouraged.

---

> > > ### Author Response · Authors · 2025-08-05
> > >
> > > We thank reviewer f1mU for the constructive and detailed comments.
> > >
> > > **One key point remains insufficiently addressed: the quantitative comparison with SALAD under equivalent conditions.**
> > > We agree that retraining SALAD on our data would be insightful, and we appreciate the reviewers' understanding that doing so is difficult within the rebuttal period. We will nevertheless do our best to include a representative baseline for the camera-ready.
> > >
> > > We, however, point out that SALAD was trained on the (illegal) GSV-Cities dataset, which is derived from Google StreetView imagery and thus offers the same data quality (albeit at a smaller scale). We also note that we did retrain multiple other baselines on our data (Fervers et al [29], SALAD-Aerial [23]) and that the proposed approach outperforms them on all metrics. We therefore believe that the main claim of our submission, around hybrid localization, still stands. Moreover, we believe that an evaluation with ANN / dimensionality reduction would not challenge the claims of our submission, as Table 9 (supplementary) already provides an upper bound with exact NN search - approximate search will only widen the gap.
> > >
> > > Finally, we highlight that the proposed hybrid strategy has benefits beyond only accuracy and scalability. Unlike pure ground-ground retrieval, it can cover areas in which there are no database ground images, as evaluated in Table 8. This makes it more widely applicable.
> > >
> > > **What is the performance when train with ArcFace/CosFace?**
> > > We are unfortunately unable to provide reliable numbers here without retraining, as we trained and evaluated these variants only early in the project with a different training dataset. We can, however, confidently state that a model trained with ArcFace/CosFace for classification is stronger than the cross-entropy baselines but weaker than our model trained with cell prototypes only.
> > >
> > > **What is the impact of the selection threshold and bidirectional loss?**
> > > The bidirectional loss between ground and aerial images resulted in +1.5% recall@1@200m on BEDENL for the SALAD-Aerial baseline. We did not perform extensive ablations on the selection threshold, but excluding it completely (i.e., no filtering) significantly reduced the Recall@1@200m on BEDENL by ~4% on the model trained with cell prototypes, with the accuracy saturating more quickly during training. We kindly note that both selection threshold and bidirectional loss were already used in Fervers et al [29].
> > >
> > > **What are some indicative results of GeoCLIP and classification-based methods with other settings?**
> > > Our best, re-trained GeoCLIP model achieved <5% Recall@1@200m on BEDENL, and was thus weaker than the best softmax-based classification baselines on this dataset (8.1% for the hierarchical loss [2] and 10.2% for the haversine loss from PIGEON [1]).
> > >
> > > It is unclear to us what “classification-based methods with other settings” refers to, since we already tuned their hyperparameters to the extent possible. We would appreciate a clarification on this point.
> > >
> > > **Moreover, in the response to another reviewer, the authors mention that a key motivation for the method is to be applied to arbitrary imagery, which contradicts the earlier justification for excluding such datasets.**
> > > We apologize for the lack of clarity in our previous response. The referenced statement in our response to Reviewer _bt81_ was about the general motivation for the problem of global geo-localization, and not specifically for our approach. The point of this statement was to highlight that geolocalization can be applied to imagery without GPS tags, and use cases where this is useful / necessary. Handling arbitrary images would also include indoor images or drone footage, which we also never claim to solve in our work. We will therefore tone down this statement.

---

> > > > ### Comment · Reviewer_f1mU · 2025-08-05
> > > >
> > > > Thanks to the authors for providing additional results and clarifications. Regarding the “classification-based methods with other settings,” the reviewer is specifically interested in the classification-based methods trained at different cell granularities (e.g., L=16), as suggested in the original review. Was the choice of cell granularity part of the tuning process? If so, can some indicative results be provided?

---

> > > > > ### Author Response · Authors · 2025-08-07
> > > > >
> > > > > We thank the reviewer for the clarification. The cell granularity is a general parameter of our experimental setup that has the same impact across baselines. It was thus not tuned per approach. We chose L15 to balance accuracy and scalability. With smaller cells, the classification performance increases, as each cell packs less visual content, but the size of the database, and thus both memory and computational cost, also increase. We analyzed this trade-off for our approach in Table 5 and observed a similar trend for classification baselines. Finer cells also increase visual aliasing which can be mitigated with a selection threshold or multiple disjoint classifiers as in [B]. Running at L17 or finer would only be possible on a smaller area but not at the scale that we consider.

---

> > > > > > ### Comment · Reviewer_f1mU · 2025-08-08
> > > > > >
> > > > > > Thanks to the authors for their reply. The reviewer identifies the above as another missing baseline and urges the authors to perform such experiments and include relevant results (if meaningful) in the final version of the paper, for a more complete comparison with the approaches from the literature.

---

### Note · Authors · 2025-08-16

We thank all reviewers for their time and valuable and constructive feedback. We summarize the main strengths and discussion points.

**Strengths:**

While all reviewers agree that the method is original, reviewer _7jVE_ notes that the method is “a conceptually simple yet powerful hybrid approach”, and _n8dV_ highlights that the “synergy overcomes the individual limitations of each approach”. _bt81_ highlights the “novel architecture and training scheme”.

The scale of our experiments was commended by all authors. Reviewer _bt81_ notes that the “train/eval split is temporal”, which is “representative” of practical “use cases”. A “meaningful evaluation of geolocation methods at a scale relevant for real-world deployment” is attested by _f1mU_, while reviewer _n8dV_ finds that this “exceeds prior work” and is a “leap from previous state-of-the-art” in scale.

“Extensive", "very complete" experiments and a “detailed quantitative analysis” are acknowledged by _n8dV_, _7jVE_ and _bt81_, respectively. _f1mU_ and _n8dV_ note that our experiments demonstrate generalization to completely new areas and data domains.

Reviewer _f1mU_ commends the model's “very high performance” and that the achieved precision at this scale is “a significant advancement”. Reviewer _7jVE_ finds the “training pipeline is interesting” and that the method tackles “a very interesting topic for the community”. _n8dV_ mentions that the “spatial error maps” and “PCA visualizations of prototypes” provide valuable insights.

**Discussion:**

To aid reproducibility, we will release the training and evaluation code and evaluate the possibility to set up an evaluation server with a hidden test set.

Although not critical to support our central claims around hybrid localization, we acknowledge that evaluating ground retrieval with SALAD on our full dataset would be interesting. Contacting the SALAD authors revealed that models with smaller dimensions do not exist anymore. Nevertheless, the evaluation in Table 9 already provides a strong, compelling comparison to the official SALAD model. We will do our best to also add a representative and fair SALAD baseline to Table 1.

We thank all reviewers for the comments on clarity and readability, and we will add the proposed explanations and an extra pipeline figure to the additional page of the final version.

Finally, we want to thank the reviewers for the insightful reviews and constructive discussion, which helped us to improve the quality of our work.

---

### Decision · Program_Chairs · 2025-09-17

**Decision:**

Accept (poster)

**Comment:**

Following the discussion phase, all reviewers recommended acceptance (1 Accept, 3 Borderline Accepts), noting that the paper addresses an important topic, is well-written, demonstrates novelty, is thoroughly evaluated, and presents strong results. The rebuttal addressed many of the reviewers’ concerns—for example, by clarifying certain technical details. However, some concerns remained regarding reproducibility, as the method was trained on private data that cannot be released due to licensing restrictions, and the model weights will also not be made available. Ultimately, all reviewers maintained their recommendation for acceptance. Accordingly, the ACs have decided to accept the paper. Please take the reviewers’ feedback into account when preparing the camera-ready version.